# Sarcopenia in the Aging Process: Pathophysiological Mechanisms, Clinical Implications, and Emerging Therapeutic Approaches

**DOI:** 10.3390/ijms262412147

**Published:** 2025-12-17

**Authors:** Larissa Parreira Araújo, Ana Clara Figueiredo Godoy, Fernanda Fortes Frota, Caroline Barbalho Lamas, Karina Quesada, Claudia Rucco Penteado Detregiachi, Adriano Cressoni Araújo, Maria Angélica Miglino, Elen Landgraf Guiguer, Rafael Santos de Argollo Haber, Eliana de Souza Bastos Mazuqueli Pereira, Virgínia Cavallari Strozze Catharin, Vitor Cavallari Strozze Catharin, Lucas Fornari Laurindo, Sandra Maria Barbalho

**Affiliations:** 1Department of Biochemistry and Pharmacology, School of Medicine, Universidade de Marília (UNIMAR), Marília 17525-902, SP, Brazil; 2Department of Gerontology, School of Gerontology, Universidade Federal de São Carlos (UFSCar), São Carlos 13565-905, SP, Brazil; 3Department of Biochemistry and Nutrition, School of Food and Technology of Marília (FATEC), Marília 17500-000, SP, Brazil; 4Graduate Program in Structural and Functional Interactions in Rehabilitation, School of Medicine, Universidade de Marília (UNIMAR), Marília 17525-902, SP, Brazil; 5Division of Cellular Growth, Hemodynamic, and Homeostasis Disorders, Graduate Program in Medical Sciences, Faculdade de Medicina, Universidade de São Paulo (USP), São Paulo 01246-903, SP, Brazil

**Keywords:** aging, muscle loss, sarcopenia, cognitive decline, neurodegenerative diseases

## Abstract

In the face of population aging, sarcopenia has emerged as a significant muscle disorder characterized by the progressive loss of muscle mass, strength, and function. Chronic inflammation, oxidative stress, and mitochondrial dysfunction contribute to sarcopenia and help explain its association with comorbidities such as type 2 diabetes, obesity, and neurodegenerative diseases. Despite extensive research, there remains a need to integrate current knowledge on interventions that target these interconnected mechanisms. This review synthesizes recent evidence on the effects of resistance exercise, nutritional supplementation (high-protein intake, leucine, vitamin D, omega-3 fatty acids), and probiotic use on muscle function and inflammatory status in older adults with sarcopenia. Literature was critically analyzed to evaluate the efficacy of multicomponent strategies. The reviewed studies consistently report that combining resistance training with anti-inflammatory nutrition and targeted supplementation improves muscle strength, reduces pro-inflammatory cytokines, and supports mitochondrial function. These findings suggest that an integrated, multicomponent approach represents a promising strategy for attenuating the progression of sarcopenia and reducing its associated comorbidities.

## 1. Introduction

Sarcopenia is defined, according to the European Working Group on Sarcopenia in Older People 2 (EWGSOP2) study, as a muscle disorder characterized by progressive and generalized loss of the quality and quantity of skeletal muscle structure [1], resulting in adverse events such as difficulty in locomotion [2] and, in extreme cases, mortality [3]. Studies show that sarcopenia currently has a prevalence of 8% to 36% in individuals under 60 years of age and 10% to 27% in those over 60 years old, demonstrating an escalating trend in terms of prevalence annually [4].

Because sarcopenia is fundamentally an age-related condition, understanding how aging alters muscle homeostasis is essential to clarifying its mechanisms. Aging acts as the biological backdrop against which these structural and metabolic changes occur. Aging disrupts the balance between anabolic and catabolic pathways, leading to structural changes such as reduced type II fiber size and number, increased fat infiltration, and fewer satellite cells, which are responsible for replacing and repairing damaged muscle fibers [5,6,7]. This decline is attributable to alterations in systemic signaling pathways, including the transforming growth factor beta (TGF-β) and myogenin, which regulate satellite cell differentiation and activation. Other factors contributing to these changes include neuromuscular junction dysfunction, loss of motor units, chronic inflammation, and insulin resistance [8,9,10,11,12].

Sarcopenia also directly reduces the quality of life because it is closely linked to chronic conditions such as type 2 diabetes mellitus (T2DM) and obesity. A large part of the population is vulnerable to sarcopenia at some point in their lives [13,14]. Patients who may develop sarcopenia include the elderly, underweight people, and people with other chronic conditions. Moreover, individuals with T2DM are more likely to be affected by this condition, as it can aggravate metabolic disorders and compromise response to treatments [15,16,17].

Sarcopenia develops from various etiologies and can be characterized as primary or secondary. Primary sarcopenia is primarily associated with age, while secondary sarcopenia is influenced by risk factors related to muscle loss, such as sarcopenic obesity, cancer, malnutrition, and rheumatic diseases [18,19]. It is also essential to highlight the risk factors for primary sarcopenia as contributors to secondary sarcopenia, as a sedentary lifestyle and physical inactivity are triggers of this condition, even in early life [20].

These combined factors show that aging not only drives sarcopenia but also contributes to a broader systemic deterioration, affecting various physiological systems beyond skeletal muscle. This connection establishes a biological continuum between muscle aging and broader degenerative processes. Thus, hormonal changes associated with aging were observed to trigger muscle loss, with androgens playing a crucial role in maintaining cellular metabolic activity by suppressing catabolic processes and exerting an anti-inflammatory effect on peripheral tissues [21]. Therefore, low testosterone levels are associated with the pathophysiology and progression of age-related diseases, such as sarcopenia, thus establishing hormone replacement therapy as a pillar in the treatment of different pathologies [22,23,24].

Increasing evidence suggests that the mechanisms leading to muscle atrophy in sarcopenia—such as chronic inflammation, mitochondrial dysfunction, and hormonal imbalance—are also implicated in neurodegenerative processes. Thus, the muscle-brain axis has emerged as a critical focus in geroscience, linking functional decline to cognitive decline. On the other hand, it is known that advanced aging is still a crucial factor in the development of dementia, which is directly associated with disability and mortality [4,25]. Other factors, such as genetic, socioeconomic, and environmental factors, as well as physical activity and a balanced diet, are key determinants of cognitive impairment [26,27]. Geroscience offers molecular explanations for the pathophysiology of sarcopenia, frailty, and cognitive decline, encompassing several age-related processes. Studies show that muscle mass can serve as a tunable biomarker in dementia prevention. In other words, neurodegenerative changes tend to emerge with advancing age and are linked to the progressive and widespread loss of skeletal muscle, known as sarcopenia [28,29,30]. Parkinson’s disease (PD), a movement disorder characterized by the death of dopaminergic neurons, is also shown to be a change related to muscle impairment [31]. Thus, studies show the prevalence of sarcopenia in more than 50% of those affected by PD, which is associated with worse progression, greater motor impairment, and high frequency of falls, in addition to affecting non-motor symptom outcomes [32,33,34].

Sarcopenia and neurodegenerative diseases currently represent a significant socioeconomic challenge, generating high healthcare costs and a loss of functional independence and productivity, significantly reducing quality of life. Furthermore, they overload public healthcare systems, thus widening social inequalities.

Because these mechanisms overlap, studying sarcopenia and neurodegeneration together becomes essential, as understanding one can illuminate aspects of the other. Therefore, this review seeks to comprehensively review the intersection between sarcopenia and neurodegenerative diseases, considering aging, muscle loss, and cognitive decline. The detailed methodology employed for this review is presented in Appendix A.

## 2. Sarcopenia, Related Risk Factors, and Therapeutic Strategies

### 2.1. Cellular Mechanisms


**Sarcopenia and Inflammation**


Several pathologies, such as T2DM, osteoarthritis, cardiovascular diseases, and cirrhosis, are associated with sarcopenia [35]. Notwithstanding, oxidative stress and inflammation become crucial from the earliest stages through the development of its pathophysiology [36,37,38]. Inflammation is defined as a series of tissue responses to injury that may be associated with various pathologies [39,40]. Some authors indicate that inflammation impacts skeletal muscle mass, strength, and function. Current studies offer new insights into the molecular underpinnings of sarcopenia, specifically through a more specific inflammatory mechanism, pyroptosis, which involves apoptotic cells characterized by programmed cell death and is associated with inflammatory processes [41,42,43,44,45]. Pyroptosis is induced through activation of the Nucleotide-binding domain, leucine-rich repeat pyrin domain-containing protein 3 (NLRP3) inflammasome. Preclinical studies, particularly in animal models of denervation-induced muscle atrophy, indicate that NLRP3 activation stimulates the ubiquitin-proteasome system (UPS), promoting protein breakdown and subsequent muscle atrophy [46]. Additional evidence from bioinformatic analyses and mechanistic reviews supports a potential link between inflammation, pyroptosis, and sarcopenia, although direct evidence in humans remains limited [42,47].

Evidence indicates that aging-related increases in reactive oxygen species (ROS) impair mitochondrial function and muscle nutrient handling, promote protein oxidation and degradation, and lead to muscle loss. Elevated levels of inflammatory biomarkers, such as C-reactive protein (CRP), interleukin-6 (IL-6), and tumor necrosis factor alpha (TNF-α), are associated with muscle loss and decreased strength in the elderly [37,38,48,49]. In addition, the high number of surrounding pro-inflammatory cytokines, characteristic of chronic low-grade inflammation, increases TNF-α, IL-6, and interleukin-1β (IL-1β), further favoring anabolic resistance and the loss of strength, muscle mass, and functionality of skeletal muscles [38].

Studies show that inflammation also interacts with immune cells, as these are linked to the development of sarcopenia. Leukocytes, including neutrophils, lymphocytes, and monocytes, modulate inflammation. Notably, monocytes are more closely associated with an increased risk of cardiovascular mortality in sarcopenic patients [50,51,52]. Leukocytes, on the other hand, exhibit immunological activity that increases oxidative stress and, consequently, the onset of sarcopenia by releasing cytokines and damaging muscle fibers through ROS [52].

Inflammation also disrupts muscle metabolism, contributing to insulin resistance and metabolic syndrome—both key risk factors for sarcopenia. Inflammation can also impede muscle protein synthesis and function, leading to skeletal muscle weakness and atrophy, which is closely linked to the onset of syndromes such as frailty. Muscle mass loss is a key factor in its onset, affecting quality of life through symptoms such as unintentional weight loss, exhaustion, weakness, and poor physical performance [52,53]. Therefore, it is noted that inflammation has a close relationship with the development of sarcopenia and is present in several chronic pathologies; that is, it is a gateway for the appearance of risk factors, such as malnutrition, obesity, anabolic resistance, and mitochondrial damage, which can induce sarcopenia, especially in older individuals [54,55].


**Sarcopenia and Oxidative Stress**


Oxidative stress is characterized by an imbalance in the production of ROS, which can damage the skeletal muscle genome. Increased oxidative stress in elderly patients is recognized as one of several risk factors for the development of sarcopenia [56,57,58]. Studies have shown that increased oxidative damage to proteins, lipids, and genetic material in skeletal muscle occurs with aging, as evidenced by increased lipid peroxidation, protein carbonylation, and deoxyribonucleic acid (DNA) damage [59]. Therefore, it is well established that oxidative stress plays a crucial role in age-related muscle deterioration. The age-related decline in the antioxidant capacity, combined with increased ROS production, contributes to impaired satellite cell function and reduced regenerative capacity. This oxidative damage is more prevalent in older men than in women [49,60,61,62].

Superoxide anion and nitric oxide are the primary free radicals present in muscle fibers, affecting the mitochondria, sarcoplasmic reticulum, sarcolemma, cytosol, and transverse tubules [62]. Among specific cellular structures, mitochondria are the primary targets of oxidative attack, as they are the main source of ROS in cells, due to their close association with the source of oxidants, their lack of histones and introns, and their less intensive repair compared to nuclear DNA [63]. Furthermore, studies on organelle aging demonstrate that increased oxidative stress in the elderly is primarily responsible for cellular damage [56]. Oxidative damage to mitochondria and proteins can also result in mitochondrial dysfunction [64,65].

Studies have shown that the decrease in muscle regeneration capacity in sarcopenia is due to oxidative damage in myoblasts, which play a crucial role in muscle regeneration by undergoing myogenic differentiation to fuse and restore damaged muscle; this process is also impaired by aging [57]. Furthermore, it is well established that increased ROS during aging coincides with the pro-oxidant effect of physical inactivity, leading to a significant increase in oxidative stress, as expected in aging muscle. Studies have shown that elevated biomarker levels were present in both resting and exercising muscles of elderly individuals [59]. The mechanisms of antioxidant defense in these older adults are still inconclusive. On the other hand, there is evidence of a decrease in the enzymatic antioxidant system in the muscles of this group, as demonstrated by lower catalase and glutathione transferase activities during senescence [66]. However, it becomes evident that although physical exercise does not completely alleviate the signs of mitochondrial aging, it has beneficial effects in combating mitochondrial dysfunction caused by oxidative stress associated with advanced aging, thus highlighting its importance in the daily lives of this population [67].


**Sarcopenia and Mitochondrial Dysfunction**


Mitochondria are dynamic organelles that form an interconnected network that continuously undergoes fusion and fission, processes that maintain proper functioning and prevent the accumulation of damaged organelles [68,69]. The interaction between mitochondrial biogenesis and mitophagy degradation maintains healthy muscle. However, its disruption results in changes in cellular bioenergetics and consequent loss of muscle mass and function [70,71,72]. In other words, mitochondria serve as a crucial powerhouse in muscle fibers, so that their inability or ineffective function results in muscle impairment, characterized by mitochondrial swelling, loss of cristae, destruction of the inner membrane, and impaired respiration [73,74].

Mitochondrial dysfunction is defined as an imbalance in mitochondrial dynamics and the interruption of mitophagy. This process degrades damaged mitochondria via the lysosomal pathway, ensuring the functional health and metabolic homeostasis of cellular structures [75,76]. Additionally, the accumulation of mutations and deletions in mitochondrial DNA, along with reduced maximum O_2_ uptake, contributes to defective energy production and increased ROS generation. Excess ROS can promote degradation of myofibrillar proteins and inhibit protein synthesis. Furthermore, the pathophysiological underlying muscle fiber atrophy resulting from mitochondrial dysfunction remains to be fully elucidated [73,77,78,79,80].

Studies also show that bioenergetics is a fundamental pillar to homeostasis, since the response to mitochondrial metabolic demand declines with age, resulting in reduced effectiveness, impaired mitochondrial turnover, and altered redox homeostasis [78,81]. Thus, senescent muscle cells present significant morphological and functional changes when evaluated [82]. Therefore, mitochondrial integrity loss is considered a factor associated with muscle impairment (Figure 1). The need for therapeutic strategies that intervene in this metabolic process—specifically, mitochondrial quality-control mechanisms such as mitochondrial dynamics and mitophagy—is crucial for maintaining metabolic balance and mitochondrial health in skeletal muscle [77,83]. The interaction of mitochondrial dysfunction with other systemic processes, such as inflammation, hormonal decline, and intestinal dysbiosis, may further amplify the pathophysiological cycle underlying sarcopenia and its neurological consequences [84,85,86,87,88]. These interconnected mechanisms are illustrated in Figure 2, which integrates the inflammation–mitochondria–muscle axis with gut and neural pathways involved in aging-related functional decline.

Table 1 summarizes the key factors associated with sarcopenia, and Table 2 presents promising biomarkers for therapeutic approaches to sarcopenia.

Collectively, these mechanisms can be conceptualized as components of a unified systemic inflammation–oxidative stress–energy metabolism axis. Chronic low-grade inflammation acts as the initiating event, promoting the release of pro-inflammatory cytokines (e.g., TNF-α, IL-6, IL-1β) that activate intracellular signaling cascades leading to mitochondrial dysfunction and excessive ROS production. The resulting oxidative stress further amplifies inflammatory signaling, disrupts mitochondrial biogenesis, and impairs energy metabolism, ultimately leading to impaired muscle protein turnover and anabolic resistance. Thus, inflammation, oxidative stress, and mitochondrial dysfunction form a self-perpetuating feedback loop that drives the progression of sarcopenia. This hierarchical interplay is illustrated in Figure 1, which integrates these mechanisms into a coherent pathophysiological framework.

### 2.2. Associated Clinical Conditions


**Sarcopenia and Diabetes Mellitus**


T2DM is a complex metabolic disease, among the most prevalent worldwide. With high morbidity and mortality rates, this condition affects approximately 25% of the population over 65 years. Affected individuals have a twice-increased risk of developing sarcopenia, with an increased functional impact resulting from this loss of muscle mass, which can be secondary to T2DM, a medical condition, or aging itself. Studies have shown that this interaction is due to the negative impact of muscle mass loss in malnourished elderly diabetic patients or even the risk of malnutrition [115,116,117,118,119,120].

Analyses have shown different mechanisms to explain the greater incidence of sarcopenia in the diabetic population. Hyperinsulinemia, present in each metabolic profile, resulting from prolonged physiological increases, compromises insulin sensitivity. Consequently, the anabolic effect on skeletal muscle becomes increasingly imperceptible, leading to reduced protein synthesis and increased muscle degradation due to reduced insulin action, compromising muscle mass and strength. The catabolic state is also involved in this relationship through protein catabolism and muscle loss [115,121,122,123,124].

Furthermore, hyperglycemia is associated with several metabolic abnormalities that correlate with muscle damage and mitochondrial dysfunction, resulting in the accumulation of intramyocellular lipid metabolites. Chronic hyperglycemia promotes the accumulation of advanced glycation end products (AGEs) in skeletal muscle, and this relationship is associated with reduced grip strength, leg extension power, and slow walking speed [124,125,126,127].

Furthermore, diabetes is known to be intrinsically linked to increased levels of inflammatory cytokines, such as IL-6 and TNF-α, which contribute to muscle loss [128]. Thus, elevated levels of specific inflammatory markers are associated with decreased muscle strength and function. The protein catabolic pathway, already mentioned in this topic, can be triggered by the inflammatory cascade, leading to muscle atrophy, which in turn reduces both physical performance and functional capacity, highlighting the complexities faced by individuals with diabetes [126,129,130].

Recent evidence indicates that the relationship between sarcopenia and T2DM is bidirectional [131]. Not only does diabetes increase the risk of sarcopenia through hyperglycemia, insulin resistance, and chronic inflammation, but sarcopenia itself may exacerbate insulin resistance and impair glucose homeostasis [132]. Longitudinal studies have shown that reduced muscle mass is a predictor of diabetes development in patients with impaired glucose regulation [133]. Additionally, Mendelian randomization analyses suggest a potential causal role of sarcopenia-related traits in increasing the risk of diabetes, highlighting the complex interplay between muscle loss and metabolic dysfunction [134]. Therefore, interventions targeting muscle mass and function may have beneficial effects not only on physical performance but also on glycemic control and metabolic health.

In individuals who fit the profile of sarcopenia and T2DM, there is a concern about the reduction in bone mass (Figure 3), since it contributes to a greater risk of fractures due to bone fragility. This condition is more common in women, who are more susceptible to osteoporosis, which can end up reducing the mobility and quality of life of these people [135,136]. This increased susceptibility in women may be partly explained by hormonal changes, particularly the decline in estrogen levels after menopause, which accelerates bone resorption and contributes to both reduced bone density and loss of muscle mass [137]. Estrogen also plays a regulatory role in glucose metabolism and muscle protein turnover; thus, its deficiency can exacerbate the interplay between sarcopenia and diabetes [138]. Moreover, age-related hormonal decline in both sexes, along with reduced physical activity and nutritional intake in older adults, further increases vulnerability to muscle and bone deterioration [139].

Finally, it is known that there is a relationship between the presence of macrovascular and microvascular complications, that is, nephropathy, retinopathy, and neuropathy, in diabetic and sarcopenic individuals, which together can affect muscle mass, can lead to a reduction in physical activity, can induce muscle ischemia, decrease strength, muscle mass, and the individual’s physical performance [115,140]. Therefore, it becomes essential to implement improvements to identify sarcopenia and its risk factors in patients with T2DM. Targeted therapies should promote adequate nutrition and regular physical exercise, especially resistance and aerobic exercise, to enhance muscle mass and strength, and improve physical performance in sarcopenia [141,142].


**Sarcopenia and Obesity**


The accumulation of body fat together with sarcopenia generates sarcopenic obesity, a concept that emerged in the 90s [143,144,145], which causes a decline in physical performance, increased functional decline, decreased functional capacity, favored loss of autonomy, and further increased mortality risk. These changes are more common in sarcopenic patients with obesity, as current evidence shows that sarcopenic obesity may be associated with a greater number of metabolic disorders and an increased mortality risk than either obesity or sarcopenia alone [146,147,148,149,150,151].

The development of sarcopenia and sarcopenic obesity involves complex mechanisms (Figure 3), which include environmental and genetic factors [152]. As a result, single-nucleotide polymorphisms (SNPs) are known to play an essential role in increasing individual susceptibility to sarcopenia and sarcopenic obesity [147]. It is believed that, in addition to these factors, individual variation must be considered, including the quantity and quality of skeletal muscle in each individual. That is, although older individuals are more likely to develop these pathologies, some individuals have a greater predisposition than others [146,153,154].

Patients who are obese tend to present metabolic alterations, insulin resistance, and express inflammatory mediators, related to a sedentary lifestyle, adipose tissue disorders, and chronic or acute pathologies, which together play a relevant role in the etiopathogenesis of sarcopenic obesity [155]. As pointed out above, several risk factors can lead to the development of sarcopenia, including excess alcohol consumption, smoking, physical inactivity, processed meats, and excess salt, which are linked to an increased risk of muscle loss [153]. These factors contribute to increased fat mass, decreased muscle mass, and reduced bone mass among the elderly, which can lead to obesity, sarcopenia, and osteoporosis [145].

In summary, individuals with sarcopenic obesity are known to have a higher prevalence of developing cardiovascular problems and metabolic syndrome [156], which are strongly associated with inflammatory markers. Studies show that hyperglycemia, insulin resistance, and excess cytokine production can contribute to sarcopenia, leading to reduced fiber diameter, metabolic degradation of skeletal muscle, and a more severe sarcopenic process [44,157]. Hence, it is worth emphasizing that a balanced diet and physical activity are essential as prophylactic treatments for sarcopenia and obesity, as they aim to reduce the onset of several risk factors for these pathologies, both individually and in combination. In addition, it is necessary to control conditions such as smoking, alcohol consumption, and a sedentary lifestyle [150,158,159,160].


**Sarcopenia and Neurodegenerative Diseases**


Sarcopenia is associated with several adverse health consequences, including the risk of falls, fractures, physical disability, and increased mortality. However, beyond these factors, attention has focused on the relationship between sarcopenia and cognitive impairment, including Alzheimer’s disease (AD). This prevalent degenerative brain disease is the leading cause of dementia worldwide and is associated with impaired cognitive function, which, like sarcopenia, tends to develop with aging. Studies have shown that patients with mild cognitive impairment (MCI), AD, and PD have a higher prevalence of sarcopenia, which can lead to worsening physical and cognitive health [161,162,163,164].

Even in the early stages of AD and throughout its progression, it is well established that physical performance, particularly gait, muscle strength, and balance, deteriorates [165,166]. Despite the evidence that sarcopenia is closely associated with cognitive impairment, there is currently no conclusive mechanism. Therefore, it is known that sarcopenia and AD share similarities, such as inflammation, oxidative stress, nutrition, immobility, and hormonal dysregulation. Inflammatory dysregulation can affect the central nervous system and contribute to the pathophysiological mechanisms of neurodegenerative diseases such as AD [167,168].

On the other hand, there is also a link between sarcopenia and PD, a neurodegenerative movement disorder characterized by the death of dopaminergic neurons. As the disease progresses, various symptoms emerge, including reduced motor skills, body changes, and decreased physical performance, which can affect the vitality of these patients. These same signs and symptoms have been observed in patients who develop sarcopenia after aging, demonstrating a strong link between aging and the interaction between sarcopenia and neurodegenerative diseases [31,169,170]. Studies have shown that the interaction between sarcopenia and PD is related to their common pathway, neuroinflammation, with elevated levels of inflammatory mediators detected in sarcopenic and PD patients. Furthermore, muscle loss and poor physical performance in PD patients have been associated with elevated levels of IL-6. Moreover, mild motor neuron degeneration is another mechanism of neurogenic sarcopenia in PD, with a low number of motor units observed in these patients. Sarcopenia can be influenced by hormonal changes, with androgens being essential for maintaining muscle mass. Low testosterone levels can also cause or accelerate age-related diseases such as sarcopenia [171,172,173].

It is well known that declines in cognitive and physical performance are common in old age. Geroscience, the scientific field responsible for the study of biological aging and the phenomena it triggers, has supported the association regarding the pathophysiology behind physical functional decline, cognitive decline, and sarcopenia, which share physiological pathways and processes that culminate in these diseases [74,174,175,176,177].

Among the factors contributing to the development of these pathologies, particularly in the context of cognitive decline, are mitochondrial dysfunction, alterations in cellular organization, changes in endocellular signaling, inflammation, and metabolic shifts, all of which are also observed in muscular senescence [178]. Thus, the presence or absence of sarcopenia in the elderly may be a crucial factor in cognitive performance [179,180].

On the other hand, neurological factors are among the determinants of sarcopenia development, as neuromuscular junctions—synapses that connect the skeletal muscle system to the nervous system—play a crucial role in aging-related musculoskeletal disorders. Specific patterns of circulating metabolic and neurotrophic factors have been associated with the development of physical functional decline and sarcopenia. These factors are primarily linked to changes in the conversion of proteins into energy and an insufficient intake of calories and proteins necessary for maintaining muscle mass. Some studies have also described the association between sarcopenia and cognitive decline in the elderly, which is likely mediated by myokines, substances produced by muscles that participate in communication between the brain and skeletal striated tissue [74,109,181,182,183,184]. Figure 4 shows the relationship between sarcopenia and neurodegenerative diseases.

Furthermore, studies reveal that sarcopenia and cognitive decline share risk factors, such as diabetes, systemic arterial hypertension, and cerebrovascular diseases. Chronic inflammation resulting from immuno-senescence and increased cytokine release has been shown to have harmful effects. Elevated levels of IL-6, CRP, and TNF-α have been investigated as possible pathophysiological mechanisms involved in sarcopenia. Dysregulation of the inflammatory response may also contribute to the development of MCI. Furthermore, studies suggest that adopting a Mediterranean diet, combined with a high intake of quality proteins, may reduce the risk of MCI and sarcopenia. Consequently, the balance between muscle protein synthesis and degradation levels is an essential factor in the etiology of muscle atrophy in individuals with sarcopenia. Thus, inadequate nutritional intake may be one of the possible explanations for the association of the manifestation of MCI together with sarcopenia in the elderly; consequently, interventions in sarcopenia or cognitive decline may bring numerous long-term benefits to patients affected by these pathologies [185,186,187,188,189].


**Sarcopenia and Aging**


Sarcopenia is associated with aging, as cellular senescence and subsequent metabolic changes lead to a progressive decline of skeletal muscle mass and function [190]. Thus, there is a rise in the prevalence of sarcopenia with advancing age. Elderly adults aged 70 and above experience a 20% loss of muscle mass [191]. However, this condition reflects the cumulative effects of a lifetime, influenced by lifestyle habits, physical exercise, and genetic factors, resulting in a decline in strength capacity in adult individuals from the age of 30 [192,193].

The reduction in anabolic components, such as growth hormone (GH) and insulin-like growth factor one (IGF-1), associated with the greater expression of catabolic substances that occurs with aging, compromises muscle homeostasis, thus reflecting a pathogenic pillar for sarcopenia [194,195]. Anabolic resistance is therefore established in the elderly as an attenuated response to physical and nutritional stimuli, leading to reduced muscle protein synthesis in the presence of naturally occurring hormonal dysregulation [196]. Furthermore, the subclinical inflammatory state in elderly individuals contributes to the establishment of the pathogenic process, as evidenced by elevated levels of inflammatory cytokines [197,198].

Due to the lower muscle index, affected individuals have a greater risk of fractures due to fragility, falls, cognitive disorders, and numerous geriatric syndromes, thus compromising individual autonomy [199]. Based on this, it is possible to conclude that the most effective management for sarcopenia is a multi-domain intervention that targets physical, cognitive, and nutritional health, aiming to increase the population’s life expectancy [200,201]. It is also necessary to reduce the pro-inflammatory state in this population group through physical exercise, preferably resistance training, in combination with long-term nutritional support that emphasizes protein intake [202,203]. 

### 2.3. Gut Sarcopenia-Cognition Interaction

Homeostasis is established through various mechanisms, including bidirectional communication between the brain and the intestinal microbiota, mediated by endocrine, neural, or immunological factors [204,205]. Therefore, it is considered that during aging, it is natural for transitions to happen in the composition of the intestinal microbiota due to changes in colonization patterns, which may, in turn, influence the inflammatory state, as well as the development of neural metabolic changes and age-related pathophysiological processes [206,207,208].

Dysbiosis, therefore, appears to be a key factor in the dysregulation of the brain–gut axis, resulting in impaired neurodevelopment, gastrointestinal disorders, and physiological changes in the inflammatory process of the central nervous system [209,210]. Thus, the activation of neuroinflammatory pathways, such as the NLRP3 inflammasome, is directly associated with the diagnosis of neurodegenerative pathologies, including AD, PD, and Multiple Sclerosis [211,212,213].

The gut–brain axis also affects skeletal muscles, forming the microbiota-gut-muscle axis [214,215]. This demonstrates the presence of a low-grade inflammatory state and anabolic resistance, due to changes in the composition of the intestinal microbiota, which may favor the development of muscular phenotypes such as sarcopenia in already frail elderly individuals, by compromising the availability of lean muscle mass and affecting the function of a specific cell group [216,217,218].

Recent studies have provided more profound insight into the mechanistic pathways linking gut dysbiosis to both sarcopenia and cognitive decline. A reduction in short-chain fatty acid (SCFA)–producing bacteria, particularly *Faecalibacterium prausnitzii* and *Roseburia* spp., leads to decreased levels of butyrate [219]. This key metabolite maintains intestinal barrier integrity and exerts anti-inflammatory and neuroprotective effects [220]. Loss of this metabolite increases intestinal permeability, facilitating the translocation of lipopolysaccharides (LPS) into the circulation, thereby activating Toll-like Receptor 4 (TLR4)/Nuclear Factor Kappa B (NF-κB) signaling and promoting inflammation [221,222]. This chronic inflammatory state not only triggers microglial activation and NLRP3 inflammasome assembly in the brain but also impairs skeletal muscle homeostasis by disrupting insulin–protein kinase B (Akt) signaling, impairing mitochondrial function, and enhancing proteolytic activity via the ubiquitin–proteasome and autophagy–lysosomal systems [86,223,224,225,226]. Additionally, alterations in the tryptophan–kynurenine pathway have been implicated in both neurodegenerative and sarcopenic processes [227]. Together, these mechanisms establish a shared inflammatory–metabolic network—the gut–muscle–brain axis—linking microbial imbalance to the dual decline in physical and cognitive function.

Therefore, the pro-inflammatory environment resulting from aging and the consequent disturbance of the intestinal microbiota are common conditions associated with the neuro- and muscular degenerative pathological process [228,229]. Thus, early physical and nutritional interventions, such as the Mediterranean diet, appear to be positively associated with improvements in the intestinal microbiota, due to greater SCFA synthesis and a consequent favorable prognosis for frailty and neurodegenerative conditions [230,231,232]. Figure 5 summarizes the relationship among dysbiosis, neurodegeneration, and muscle loss.

### 2.4. Sarcopenia and Therapeutic Strategies

Therapies evaluated for preventing and treating sarcopenia have become widely studied, aiming to promote a better quality of life and reduce geriatric syndromes in the context of aging. Among the mechanisms considered for the treatment and prevention of sarcopenia in the elderly is dynamic resistance exercise (DRE), which, when performed at low volume/high intensity training (HIT) and combined with a protein diet, vitamin D, and calcium, proved to be significant in the scenario evaluated [233,234,235,236]. Analyses also show that high-intensity resistance training (HRT) is superior to low-load resistance exercise (LRT) for improving muscle mass, increasing muscle hypertrophy, and consequently improving the pathological state of sarcopenia. However, the high mechanical loads associated with its practice pose a barrier to its demand, leading to the search for safer approaches for vulnerable populations with greater degrees of physical disability, such as blood flow restriction (BFR) training [237].

Nutrition is a modifiable lifestyle factor in sarcopenia, which can be used to promote healthier, possibly more active aging, characterized by a continuous and gradual process of natural changes [238]. Given this, a higher protein intake is recommended to preserve muscle mass and physical fitness in old age, thereby improving physical performance and strength [238,239]. This indication is based on the fact that the muscle requires a greater quantity of amino acids to achieve maximum stimulation of protein synthesis with age, which is mainly driven by hyperaminoacidemia [238,240,241].

Failure to properly stimulate muscle protein synthesis leads to a gradual loss of muscle mass, particularly in type II fibers, which compromises muscle strength and the individual’s physical condition. However, studies have shown that the increase in type II fibers can be stimulated by physical exercise, especially resistance exercise, thereby reversing the decline in satellite cells associated with sarcopenia [242,243]. Some authors have also demonstrated that older adults with lower protein intake are more susceptible to sarcopenia than those with a protein-based diet. Furthermore, consumption of plant-based protein was associated with a lower prevalence of the disease. At the same time, older adults who consume a high-fat diet are at a higher risk, even if their protein intake is higher. It has been widely discussed that adequate protein intake can help prevent the development of sarcopenia or at least its progression, but the findings are still inconclusive [238,244,245].

The relationship between diet and maintaining muscle mass for healthy aging is currently a widely discussed topic. It has attracted the attention of researchers, particularly regarding the effects of essential nutrients, such as amino acids, vitamins, omega-3 fatty acids, and calcium, in preventing and even delaying the progression of sarcopenia when consumed in appropriate amounts and forms [246,247]. Proteins from whey have demonstrated an essential role in anabolic stimulation due to their faster digestion, resulting in a slight increase in plasma amino acid levels, in addition to the high content of essential amino acids, including leucine, which stimulates anabolism autonomously, thus demonstrating their role in reducing the risk of developing sarcopenia [248,249].

Furthermore, vitamin D supplementation has been found to improve muscle strength, particularly in elderly individuals with low serum vitamin D levels [248,250]. Studies have shown that vitamin D is essential for skeletal muscle health. It affects skeletal muscle function by regulating calcium and phosphorus metabolism, promoting muscle protein synthesis, and modulating cell proliferation and differentiation. Since this vitamin exerts synergistic effects on the protein anabolic mechanism, its deficiency is a risk factor for the development of sarcopenia. Furthermore, the combination of milk protein, leucine, and vitamin D supplementation has been shown to increase muscle mass in patients with sarcopenia. Furthermore, physical exercise combined with supplementation can improve fitness, function, and muscle gain [251,252,253,254,255].

Given these issues, the frequent consumption of ultra-processed foods increases the risk of developing sarcopenia or even precedes its onset. Processed foods often promote lower intake of nutrients, including protein, dietary fiber, vitamins A, C, and E, zinc, selenium, magnesium, and iron. Thus, obesity caused by this diet promotes muscle loss. Besides being low in antioxidants, it increases the risk of inflammation and sarcopenia [256,257]. Therefore, it is necessary to raise awareness of eating habits to reduce consumption of ultra-processed foods, especially among the older population, and thereby mitigate the development of sarcopenia [258].

Furthermore, leucine administration has been considered for the treatment of sarcopenia in the elderly, proving effective in its isolated form and as a co-supplement. In its isolated form, efficacy was demonstrated, as this amino acid was more efficient in stimulating muscle synthesis and reducing turnover. On the other hand, it was observed that when leucine is administered as a co-supplement with vitamin D, medium-chain triglycerides, and whey protein, it provides greater benefit for muscle strength. When combined, physical activity increases lean mass and strength and enhances the well-being of sarcopenic elderly individuals, compared to its isolated form [259,260,261,262,263,264]. Dietary interventions, such as amino acid supplementation, such as leucine, omega-3 fatty acids, and the probiotic *Lactobacillus*, also show significant therapeutic effects in reducing the progression of sarcopenia, acting on the physiology of muscle metabolism, regulating the turnover of muscle proteins, in addition to modulating the intestinal microbiota through its action on the muscle-intestinal axis [263,265,266].

Metabolic regulators, such as Bimagrumab^®^, are advancing in studies and remain promising in the evaluated scenario [267]. The monoclonal antibody targets activin type II receptors (ActRII), thereby regulating muscle metabolic pathways by inhibiting myostatin and other regulators of skeletal muscle mass [268,269,270]. Although selective androgen receptor modulators (SARMs) and myostatin inhibitors have been explored as potential therapeutic approaches for sarcopenia, clinical evidence remains inconsistent across randomized controlled trials, and neither agent has received regulatory approval. Therefore, these interventions are currently considered experimental and are not recommended for routine clinical use.

Among possible dietary supplements capable of modifying muscle status, the combination of hydroxymethylbutyrate, carnosine, magnesium, butyrate, and lactoferrin is also promising for treating sarcopenia, based on its effects on intestinal regulation, antioxidant effects, and direct positive action on muscle function. Thus, in validated studies, reductions in inflammatory markers were observed, along with an increase in muscle percentage in previously evaluated individuals [271,272,273,274]. Finally, multifactorial intervention is considered essential for the treatment and prevention of sarcopenia. Physical and behavioral modifications, combined with dietary supplementation, an anti-inflammatory diet, probiotics, and a protein diet, are fundamental pillars for improved muscle metabolism. Their outcomes are also being studied and evaluated in different combinations [275,276]. Figure 6 and Table 3 summarize the possible therapeutic approaches for sarcopenia.

## 3. Conclusions

Given the numerous mechanisms that contribute to sarcopenia in aging, further research is needed to elucidate its underlying mechanisms and pathological processes, and to identify effective treatments and preventive measures for clinical practice. Several pathophysiological mechanisms can cause sarcopenia. Despite the inconclusive findings, its association with inflammatory pathways, mitochondrial dysfunction, and oxidative stress is evident. Elevated levels of inflammatory cytokines, such as IL-6, TNF-α, and CRP, were also observed in comorbidities related to the development of sarcopenia, such as T2DM, obesity, mitochondrial dysfunction, and metabolic alterations, confirming the current state of occurrence in the scenario evaluated, in addition to establishing common pathways for its pathological origin. Thus, unique comorbidities, such as T2DM and AD, were shown to be associated, underscoring the need to consider shared pathological biomarkers. Furthermore, the gut–brain–muscle axis was evaluated for its influence on skeletal muscle, revealing a low-grade inflammatory state and anabolic resistance, driven by changes in the gut microbiota composition. Consequently, physical and nutritional interventions, such as the Mediterranean diet, have been shown to positively impact the intestinal environment, highlighting the need for further studies on specific probiotics and microbiota-modulating diets to establish their effectiveness in clinical practice. Furthermore, studies demonstrate the prevention of sarcopenia through regular exercise and a healthy diet. Regarding the treatment of the evaluated condition, exposure to leucine alone or in combination with vitamin D and whey protein is significantly adequate. However, clinical guidelines and therapeutic plans remain inconclusive in the medical community. Finally, promising strategies for addressing the prospects of sarcopenia include amino acid supplementation, such as leucine, omega-3 fatty acids, and the probiotic *Lactobacillus*, as well as the monoclonal antibody Bimagrumab^®^, which has demonstrated effects in reducing muscle loss through various mechanisms.

While this review provides a comprehensive synthesis of current literature on the interactions among gut microbiota, sarcopenia, and cognition, several limitations should be acknowledged. First, our study is a narrative review rather than a formal meta-analysis, and we did not conduct a systematic critical appraisal of study quality, which may influence the robustness of the conclusions. Second, the selection of included studies may be subject to publication bias, as negative or null findings are less likely to be published. Third, much of the existing literature is derived from Western populations, which may limit the generalizability of our findings to other ethnic or geographical groups. Finally, many mechanistic pathways discussed remain based on preclinical or associative data, highlighting the need for further experimental and longitudinal human studies to confirm causal relationships. Despite these limitations, we believe that this review provides a valuable framework for understanding the gut–muscle–brain axis and identifies key avenues for future research.

## 4. Future Perspectives

Given the relevance of sarcopenia and its multisystemic effects, such as cognitive decline, it is necessary to perform more studies aiming to clearly and directly demonstrate the mechanisms by which the onset of sarcopenia influences the development of various pathologies, including neurodegenerative diseases. These are also influenced by population aging itself and the most common pathologies in the elderly, such as T2DM and even obesity, all pathways linking sarcopenia to adverse health outcomes, particularly in populations with comorbidities such as cardiovascular disease, metabolic disorders, and chronic inflammation. Longitudinal studies with larger and more diverse cohorts are needed to establish predictive biomarkers and validate diagnostic criteria across different age groups and ethnicities. Furthermore, integrating multiomics approaches (genomics, proteomics, metabolomics) with advanced imaging techniques can provide deeper insights into muscle metabolism and its systemic interactions. The development and validation of targeted interventions combining nutritional strategies, physical activity protocols, and pharmacological agents should be prioritized, with a focus on individualized, precision-based medicine. Ultimately, collaborative, multidisciplinary efforts will be essential to translate mechanistic discoveries into effective public health strategies for the prevention and treatment of sarcopenia.

## Figures and Tables

**Figure 1 ijms-26-12147-f001:**
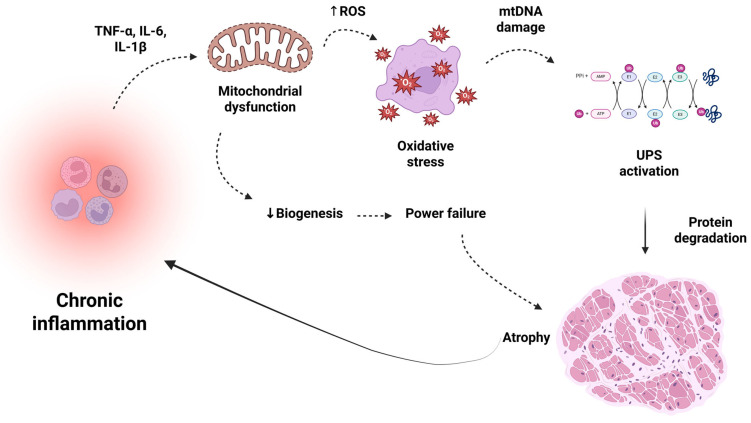
Inflammation-Mitochondria-Muscle Axis. Chronic inflammation in the elderly, mediated by pro-inflammatory cytokines such as TNF-α, IL-6, and IL-1β, leads to mitochondrial dysfunction and increased ROS production, causing oxidative stress, mtDNA damage, and UPS activation. This process can lead to protein degradation, reduced biogenesis, and impaired energy production, ultimately culminating in the muscle atrophy typical of sarcopenia. This vicious cycle perpetuates inflammation and exacerbates the loss of muscle function and mass. Vicious cycle: muscle atrophy releases pro-inflammatory myokines, fueling inflammation. IL: Interleukin; TNF-α: Tumor Necrosis Factor Alpha; UPS: Ubiquitin-Proteasome System; mtDNA: Mitochondrial DNA; ROS: Reactive Oxygen Species. Created in BioRender (https://BioRender.com/zww4a7z), last accessed on 3 December 2025.

**Figure 2 ijms-26-12147-f002:**
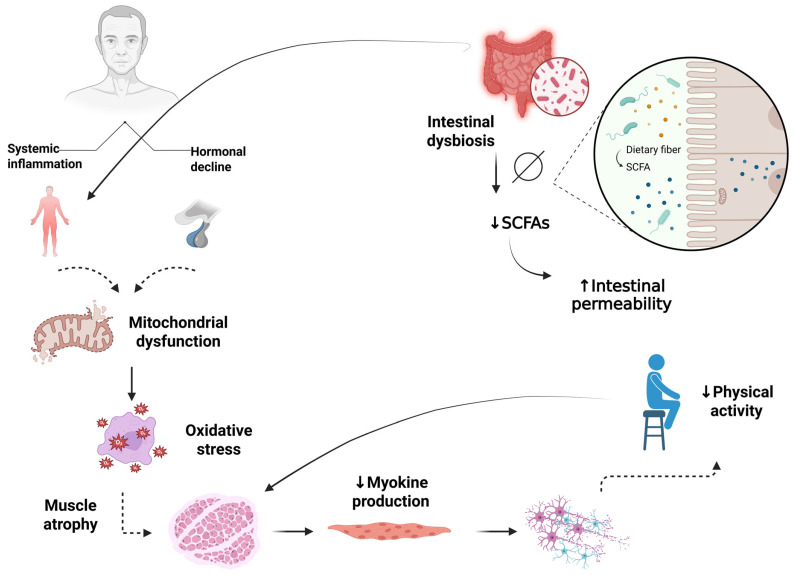
Integrated Interaction of Axes in Sarcopenia and Neurodegeneration. Aging promotes systemic inflammation and hormonal decline, leading to mitochondrial dysfunction, which in turn results in oxidative stress and muscle atrophy. Coexisting intestinal dysbiosis contributes to reduced SCFAs and increased intestinal permeability, feeding back into the inflammatory state. The resulting muscle atrophy reduces myokine production, which in turn causes neurodegeneration and a decline in physical activity, perpetuating the cycle of functional deterioration. SCFAs: Short Chain Fatty Acids. Created in BioRender (https://BioRender.com/v0emai0), last accessed on 3 December 2025.

**Figure 3 ijms-26-12147-f003:**
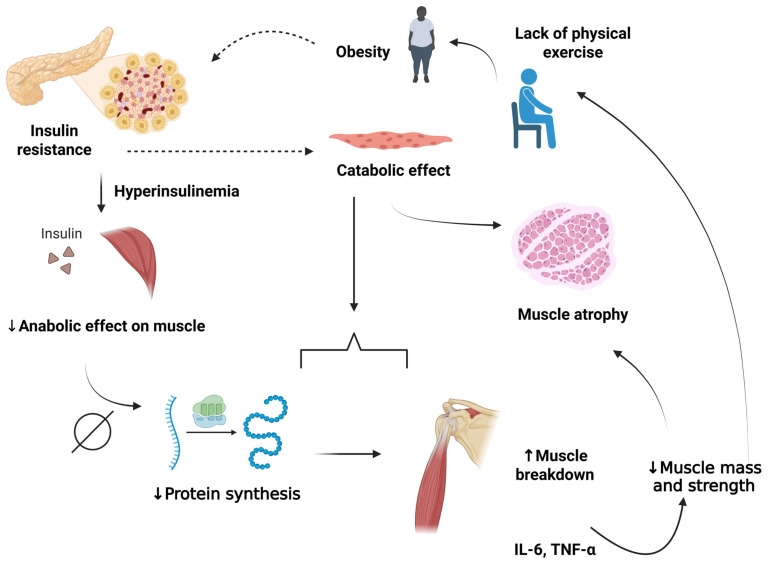
Integrated interaction between sarcopenia and diabetes mellitus. Lack of physical exercise triggers a vicious cycle: it reduces muscle mass, which is further reduced by obesity, and worsens insulin resistance, leading to hyperinsulinemia and chronic inflammation (increased IL-6 and TNF-α). This inflammation reduces protein synthesis, diminishes the anabolic effect on muscle, and accelerates muscle degradation, worsening atrophy and sarcopenia. Furthermore, insulin resistance interferes with the catabolic effect that also causes muscle atrophy. Thus, with less muscle, the motivation to exercise decreases, creating a cycle of this condition and increasing metabolic risks. IL-6: Interleukin 6; TNF-α: Tumor necrosis factor alpha. Created in BioRender (https://BioRender.com/rqc4m35), last accessed on 3 December 2025.

**Figure 4 ijms-26-12147-f004:**
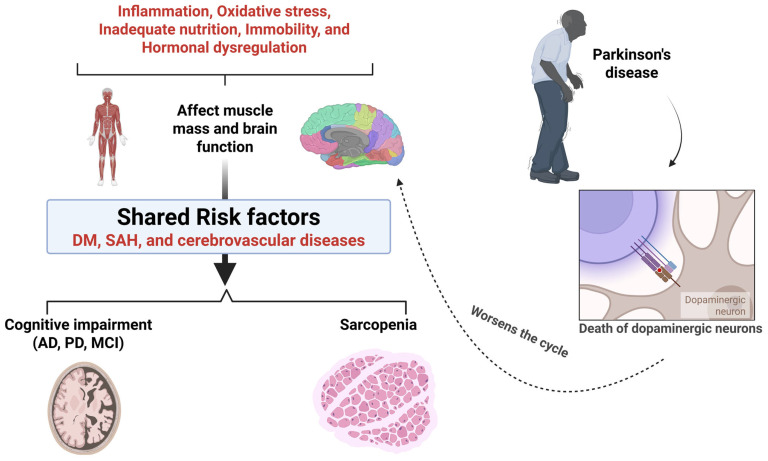
Shared pathological mechanisms linking sarcopenia, cognitive impairment, and Parkinson’s disease. Systemic factors—including inflammation, oxidative stress, inadequate nutrition, immobility, and hormonal dysregulation—negatively affect both muscle mass and brain function. These processes interact with shared risk factors such as diabetes mellitus (DM), systemic arterial hypertension (SAH), and cerebrovascular diseases, contributing to the development of sarcopenia and cognitive impairment (including Alzheimer’s disease [AD], Parkinson’s disease [PD], and mild cognitive impairment [MCI]). In PD, progressive loss of dopaminergic neurons further exacerbates motor decline, worsening immobility and perpetuating the cycle of muscle loss. The figure illustrates how these interconnected mechanisms reinforce one another, highlighting the bidirectional relationship between neurodegeneration and sarcopenia. Created in BioRender (https://BioRender.com/lqh84wu), last accessed on 3 December 2025.

**Figure 5 ijms-26-12147-f005:**
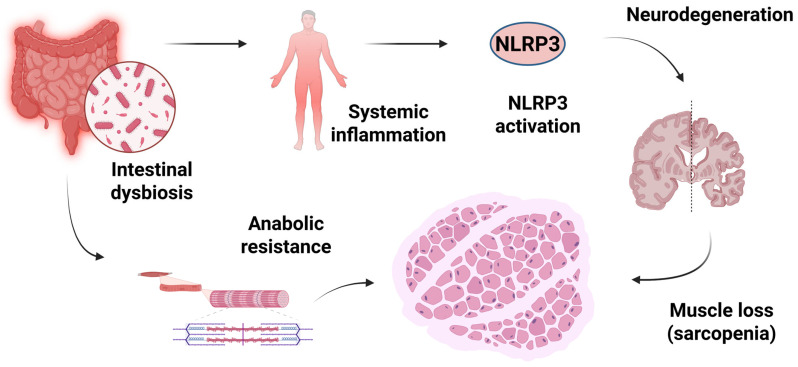
Interaction of sarcopenia with the gut–brain axis and gut-muscle axis. Dysbiosis of the intestinal microbiota triggers chronic and systemic inflammation, which affects both the brain (via the gut–brain axis) and the muscles (via the gut-muscle axis), accelerating the development/onset of sarcopenia and neurodegeneration. NLRP3: Nucleotide-binding domain, leucine-rich repeat pyrin domain-containing protein 3. Created in BioRender (https://BioRender.com/ql0gnzg), last accessed on 3 December 2025.

**Figure 6 ijms-26-12147-f006:**
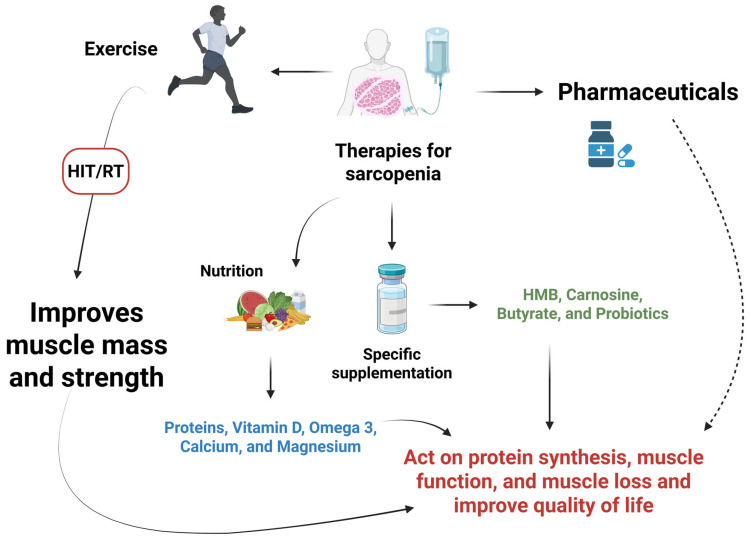
Therapies for sarcopenia. Four pillars aim to improve muscle mass and quality of life in sarcopenic patients: exercise (e.g., HIT/RT), nutrition (e.g., protein and micronutrients), specific supplementation (e.g., HMB and probiotics), and pharmacotherapy. All of these methods work together to enhance protein synthesis, improve muscle function, and reduce muscle loss, ultimately improving the quality of life for these individuals. HIT/RT: high-intensity resistance training; HMB: hydroxymethylbutyrate. Created in BioRender (https://BioRender.com/hw9w715), last accessed on 3 December 2025.

**Table 1 ijms-26-12147-t001:** Main Pathophysiological Factors of Sarcopenia.

Mechanism	Key Processes	Impact on the Muscle	References
Chronic inflammation	- ↑ TNF-α, IL-6, IL-1β; - NLRP3 inflammasome activation; - Pyroptosis.	- Proteolysis via UPS; - Anabolic resistance; - Fatty infiltration.	[89,90,91]
Oxidative Stress	- ROS accumulation; - Mitochondrial DNA damage; - ↓ Catalase/GSH activity.	- Type II fiber atrophy; - Satellite cell dysfunction; - Impaired regeneration.	[60,61,62,63,64]
Mitochondrial Dysfunction	- Impaired fusion/fission; - Mitophagy failure; - ↓ Biogenesis.	- Energy deficit; - ↑ ROS production; - Muscle apoptosis.	[83,92,93]
Anabolic Resistance	- Insulin resistance; - ↓ IGF-1/GH; - Impaired amino acid response.	- ↓ Protein synthesis; - Hypercatabolism; - Lean mass loss.	[94,95,96]
Neurodegeneration	- Neuromuscular junction degeneration; - Motor unit loss.	- Muscle denervation; - Inactivity-induced atrophy.	[97,98,99]
Hormonal changes	- ↓ Testosterone; - ↓ Vitamin D; - ↑ Cortisol.	- Reduced strength; - ↓ Muscle fiber density.	[100,101,102]

↑: Increase; ↓: Decrease; DNA: Deoxyribonucleic acid; GH: Growth hormone; GSH: Reduced glutathione; IGF-1: Insulin-like growth factor one; IL-1β: Interleukin-1β; IL-6: Interleukin-6; NLRP3: Nucleotide-binding domain, leucine-rich repeat pyrin domain-containing protein 3; ROS: Reactive oxygen species; TNF-α: Tumor necrosis factor alpha; UPS: Ubiquitin-proteasome system.

**Table 2 ijms-26-12147-t002:** Promising Biomarkers for the Therapeutic Approach of Sarcopenia.

Category	Biomarkers	Association with Sarcopenia	References
Inflammatory	- IL-6, TNF-α, CRP; - NLRP3.	- Correlated with loss of muscle mass and strength.	[37,38,48,103]
Oxidative	- Malondialdehyde; - 8-OHdG.	- Markers of accelerated muscle aging and damage.	[104,105,106]
Mitochondrial	- Cytochrome C oxidase; - mtDNA deletions.	- Indicators of energy failure and apoptosis.	[63,107,108]
Myokines	- Irisin; - Myostatin.	- Mediate muscle-brain crosstalk and regulate neuroprotection/atrophy.	[74,109]
Hormonal	- IGF-1; - 25(OH) Vitamin D.	- Low levels linked to reduced muscle mass/strength.	[110,111,112,113]
Gut dysbiosis	- SCFAs	- With aging, a decline in SCFA-producing gut microbiomes reduces SCFAs, which contribute to sarcopenia.	[114]

8-OHdG: 8-hydroxy-2-deoxyguanosine; CRP: C-Reactive Protein; IGF-1: Insulin-like growth factor one; IL-6: Interleukin-6; mtDNA: mitochondrial DNA; NLRP3: Nucleotide-binding domain, leucine-rich repeat pyrin domain-containing protein 3; SCFA: Short-chain fatty acid; TNF-α: Tumor necrosis factor alpha.

**Table 3 ijms-26-12147-t003:** Evidence-Based Interventions.

Intervention	Strategies	Evidence-Based Outcomes	References
Physical exercise	- Resistance training; - Moderate aerobic exercise.	- ↑ Muscle mass; - ↑ Strength.	[277,278,279]
Protein Supplementation	- 1.2–1.5 g/kg/day protein; - Leucine-rich sources (whey).	- ↑ Gait speed.	[280]
Vitamin D	- Supplementation (800 IU).	- ↑ Appendicular muscle mass.	[281]
Mediterranean diet	- Omega-3, MUFA, and fiber.	- ↓ Systemic inflammation; - Bone-muscular protection.	[282,283]
Gut Microbiota Modulation	- Probiotics (e.g., *Lactobacillus*); - Prebiotics.	- ↓ Muscle atrophy.	[265,266]
Multimodal Combination	- Exercise + protein + vitamin D.	- Synergistic effects (↑ muscle mass).	[263]

↑: Increase; ↓: Decrease; MUFA: Monounsaturated fatty acid.

## Data Availability

No new data were created or analyzed in this study. Data sharing does not apply to this article.

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
