# Peer review of "Sarcopenia in the Aging Process: Pathophysiological Mechanisms, Clinical Implications, and Emerging Therapeutic Approaches"

_ijms, 2025, doi:10.3390/ijms262412147_

Round 1
Reviewer 1 Report
Comments and Suggestions for Authors
The manuscript by Araujo et al., attempts to summarize the interactions between sarcopenia and various conditions including diabetes, obesity, and neurodegenerative diseases. Although the topic is of considerable interest, several important issues arise.
Major Concerns:
1. Structural and Organizational Issues
in Section 2 ("Sarcopenia and Related Risk Factors"). Subsections 2.1-2.3 address cellular mechanisms (inflammation, oxidative stress, and mitochondrial dysfunction), while subsections 2.4-2.7 discuss clinical conditions (diabetes mellitus, obesity, neurodegenerative diseases, and aging). These different topics should not be grouped under a single heading but rather reorganized into separate sections based on thematic coherence (e.g., "Cellular Mechanisms" versus "Associated Clinical Conditions").
Furthermore, despite the title's emphasis on muscle-brain crosstalk in neurodegeneration, the manuscript did not adequately focus on this relationship throughout. The content appears to address sarcopenia broadly without maintaining the promised focus on its link to neurodegeneration.
2. Inconsistencies Between Figures/Tables and Text
There are several mismatches between the figures/tables and the text that references them:
- Lines 203-204 reference Figure 1 in the context of "muscle impairment and cognition"; however, Figure 1 depicts only the "Inflammation-Mitochondria-Muscle Axis" without any representation of cognitive components.
- Lines 205-207 discuss mitochondrial quality control mechanisms and reference Figure 2, yet Figure 2 is titled "Integrated Interaction of Axes in Sarcopenia and Neurodegeneration," which does not align with the specific discussion of mitochondrial quality control.
- Table 2 (line 234) is titled "Promising Biomarkers for the therapeutic approach of sarcopenia and neurodegenerative diseases," yet contains only markers associated with sarcopenia. The biomarkers for neurodegenerative diseases are entirely absent.
Comments on the Quality of English Language
The manuscript contains many grammatical errors and inappropriate word choices that obscure meaning. Representative examples include:
- Lines 159-161: "Furthermore, the aging of these organelles showed that increased oxidative stress in the elderly is primarily responsible for cellular damage." The phrase "aging of these organelles showed" is illogical; organelles cannot "show" evidence. This should be revised to: "Furthermore, studies on organelle aging demonstrate that..." or "...have shown that..."
- Lines 172-174: "there is evidence of a decrease in the enzymatic antioxidant system present in the muscles of this group,... Hence, it becomes evident that, even though physical exercise does not completely alleviate the signs of mitochondrial aging, there are beneficial effects of physical activity in combating mitochondrial dysfunction." The transition word "Hence" is incorrectly used to introduce a contrasting statement. The text should read "However" rather than "Hence," as the sentence presents a counterpoint to the preceding statement about decreased antioxidant capacity.
- Lines 190-193: "This process seeks to degrade damaged mitochondria... However, there is the accumulation of mutations and deletions in mitochondrial DNA..." The use of "However" to connect the discussion of mitophagy and mtDNA mutations is inappropriate and creates confusion. The logical relationship between these concepts needs clarification.
Author Response
RESPONSE TO REVIEWERS' COMMENTS
Manuscript number: ijms-3908908 ― International Journal of Molecular Sciences
"Sarcopenia in the Aging Process: Pathophysiological Mechanisms, Clinical Implications, and Emerging Therapeutic Approaches"
The authors of this document wish to express their deepest gratitude to the Editor-in-Chief and the Reviewer for their thorough and insightful evaluation of our manuscript. Their expert feedback has been invaluable in enhancing the quality of our work. We have carefully considered and diligently implemented each suggestion, which has significantly improved the manuscript. We have made substantial revisions to address the points raised. These noteworthy changes are marked mainly with YELLOW-highlighted text throughout the document for ease of reference. A note will be provided for the referee's attention, highlighting corrections in a different color. Additionally, we have prepared a detailed and comprehensive response to each comment and suggestion. This response is organized in a "point-by-point" format below, ensuring that every concern has been thoroughly addressed and explained. We sincerely appreciate the time and effort invested by the Editor-in-Chief and the Reviewer, and we believe their contributions have significantly strengthened the final version of our manuscript.
REVIEWER #1
General comment
The manuscript by Araujo et al., attempts to summarize the interactions between sarcopenia and various conditions including diabetes, obesity, and neurodegenerative diseases. Although the topic is of considerable interest, several important issues arise.
General response
Dear Erudite Reviewer, thank you for taking the time to revise our manuscript and allowing us to improve based on your valuable comments and suggestions. After addressing all your comments and suggestions regarding our manuscript text, we are confident that a significantly enhanced manuscript version has emerged. We are excited to resubmit the modified version for your perusal and reevaluation. Thank you for your brilliant insights, essential contributions, and feedback. You do have an eye for improvement. As a gesture of our utmost respect for you, we would like to provide you with a detailed and comprehensive point-by-point response to your comments below. Thank you once again for your time and patience in revising our article.
Comment #1
In Section 2 ("Sarcopenia and Related Risk Factors"). Subsections 2.1-2.3 address cellular mechanisms (inflammation, oxidative stress, and mitochondrial dysfunction), while subsections 2.4-2.7 discuss clinical conditions (diabetes mellitus, obesity, neurodegenerative diseases, and aging). These different topics should not be grouped under a single heading but rather reorganized into separate sections based on thematic coherence (e.g., "Cellular Mechanisms" versus "Associated Clinical Conditions").
Response
We appreciate the reviewer’s valuable suggestion. In response, we have reorganized Section 2 to enhance thematic coherence and improve readability. The revised structure now distinguishes between cellular mechanisms and associated clinical conditions. Specifically, Section 2 is now titled “Sarcopenia and Related Risk Factors” and includes four subsections. Section 2.1 (“Cellular Mechanisms”) discusses inflammation, oxidative stress, and mitochondrial dysfunction. Section 2.2 (“Associated Clinical Conditions”) addresses diabetes mellitus, obesity, neurodegenerative diseases, and aging. Section 2.3 covers the “Gut Sarcopenia–Cognition Interaction,” and Section 2.4 focuses on “Sarcopenia and Therapeutic Strategies.” This reorganization aligns with the reviewer’s recommendation and improves the logical flow and clarity of the manuscript. The reviewer can find the corrections spanning Pages 3-17 of the revised manuscript document. We thank the reviewer for this helpful comment, which has contributed to a more transparent and cohesive presentation of the material.
Comment #2
Furthermore, despite the title's emphasis on muscle-brain crosstalk in neurodegeneration, the manuscript did not adequately focus on this relationship throughout. The content appears to address sarcopenia broadly without maintaining the promised focus on its link to neurodegeneration.
Response
We thank the reviewer for this insightful comment. We agree that the initial title placed a strong emphasis on the muscle–brain relationship, while the manuscript more broadly explores the multifactorial mechanisms of sarcopenia and its associated comorbidities. To ensure that the title accurately reflects the manuscript’s comprehensive scope, we have revised it to represent better the balance between mechanistic, clinical, and therapeutic perspectives. The revised title now emphasizes the integrative nature of the work rather than focusing solely on neurodegeneration.
Revised Title:
“Sarcopenia in the Aging Process: Pathophysiological Mechanisms, Clinical Implications, and Emerging Therapeutic Approaches” (Page 1, Lines 2-4)
This new title aligns more closely with the manuscript's content, which discusses cellular mechanisms (inflammation, oxidative stress, mitochondrial dysfunction), metabolic and neurological comorbidities (type 2 diabetes, obesity, neurodegenerative diseases), gut–muscle interactions, and evidence-based interventions. We thank the reviewer for prompting this refinement, which improves the accuracy and coherence between the title and manuscript content.
Comment #3
Lines 203-204 reference Figure 1 in the context of "muscle impairment and cognition"; however, Figure 1 depicts only the "Inflammation-Mitochondria-Muscle Axis" without any representation of cognitive components.
Response
We thank the reviewer for this careful observation. We agree that the original text inaccurately referred to cognitive aspects when describing Figure 1. To correct this, we have revised the corresponding sentence to read: “Therefore, loss of mitochondrial integrity is considered a factor associated with muscle impairment (Figure 1).” This change ensures that the figure accurately describes its content, aligning with the actual focus on the inflammation–mitochondria–muscle axis. Please find the modification in Lines 210-211 on Page 5 of the revised manuscript document.
In addition, we would like to note that all figures in the revised manuscript have been rebuilt and standardized using BioRender, ensuring more precise visualization, improved design consistency, and better alignment with the described concepts.
We thank the reviewer for the assistance in improving the precision and presentation quality of the manuscript.
Comment #4
Lines 205-207 discuss mitochondrial quality control mechanisms and reference Figure 2, yet Figure 2 is titled "Integrated Interaction of Axes in Sarcopenia and Neurodegeneration," which does not align with the specific discussion of mitochondrial quality control.
Response
We thank the reviewer for this important observation. We acknowledge that Figure 2 is a broader schematic illustrating the integrated interactions among mitochondrial dysfunction, inflammation, gut dysbiosis, and neural pathways, rather than focusing solely on mitochondrial quality control. To clarify this for the reader, we have revised the text (Lines 214-219, Page 5) to indicate that mitochondrial quality control is one component within these interconnected pathways, and that Figure 2 depicts the overall integration of these axes in sarcopenia and neurodegeneration.
Additionally, all figures have been rebuilt using BioRender to ensure clarity, consistency, and accurate representation of the mechanisms discussed.
We appreciate the reviewer’s comment, which prompted us to improve the precision of figure references and ensure alignment between figure content and the text.
Comment #5
Table 2 (line 234) is titled "Promising Biomarkers for the therapeutic approach of sarcopenia and neurodegenerative diseases," yet contains only markers associated with sarcopenia. The biomarkers for neurodegenerative diseases are entirely absent.
Response
We thank the reviewer for highlighting this critical observation. We acknowledge that the original title of Table 2 suggested coverage of biomarkers for both sarcopenia and neurodegenerative diseases, whereas the listed markers primarily relate to sarcopenia. To ensure accuracy and clarity, we have revised the title of Table 2 to “Promising Biomarkers for the Therapeutic Approach of Sarcopenia.” The revision can be noted in Line 250 on Page 7 of the revised manuscript document. We note that some of the markers included, such as irisin and myostatin, are involved in muscle–brain crosstalk and have been shown to play roles in neuroprotection; however, the primary focus of this table is on biomarkers associated explicitly with sarcopenia. This revision ensures that the table content is fully aligned with its title and the corresponding discussion in the text, thereby avoiding potential confusion for the reader.
We sincerely thank the reviewer for this constructive comment, which helped improve the clarity and precision of our manuscript.
Comment #6
Lines 159-161: "Furthermore, the aging of these organelles showed that increased oxidative stress in the elderly is primarily responsible for cellular damage." The phrase "aging of these organelles showed" is illogical; organelles cannot "show" evidence. This should be revised to: "Furthermore, studies on organelle aging demonstrate that..." or "...have shown that..."
Response
We thank the reviewer for this insightful and constructive comment. We agree that the original phrasing, “the aging of these organelles showed,” was illogical, as organelles themselves cannot “show” evidence. To improve clarity and scientific accuracy, we have revised the sentence to read: “Furthermore, studies on organelle aging demonstrate that increased oxidative stress in the elderly is primarily responsible for cellular damage” (Lines 168-170, Page 4). This change more accurately reflects the source of the evidence and appropriately attributes the findings to research studies rather than to the organelles themselves.
We believe this revision enhances the readability and precision of the manuscript, ensuring that the discussion of oxidative stress and organelle aging is both scientifically rigorous and logically coherent. We sincerely thank the reviewer for highlighting this point, as it has helped us improve the clarity and accuracy of our text.
Comment #7
Lines 172-174: "there is evidence of a decrease in the enzymatic antioxidant system present in the muscles of this group,... Hence, it becomes evident that, even though physical exercise does not completely alleviate the signs of mitochondrial aging, there are beneficial effects of physical activity in combating mitochondrial dysfunction." The transition word "Hence" is incorrectly used to introduce a contrasting statement. The text should read "However" rather than "Hence," as the sentence presents a counterpoint to the preceding statement about decreased antioxidant capacity.
Response
We thank the reviewer for this careful and constructive comment. We agree that the original use of the transition word “Hence” was not appropriate in this context, as it suggested a logical consequence rather than introducing a contrast. To improve both the clarity and the logical flow of the manuscript, we have revised the sentence to read: "However, it becomes evident that although physical exercise does not completely alleviate the signs of mitochondrial aging, it has beneficial effects in combating mitochondrial dysfunction caused by oxidative stress associated with senility, thus highlighting its importance in the daily lives of this population…" (Lines 182-186, Page 4).
This revision more accurately reflects the intended contrast between the observed decrease in the enzymatic antioxidant system and the positive effects of exercise on mitochondrial function. By using “However”, the text now clearly conveys that physical activity can provide measurable benefits despite the persistence of some age-related mitochondrial impairments. We sincerely appreciate the reviewer’s attention to this detail, which has helped us enhance the readability, logical coherence, and precision of our discussion.
Comment #8
Lines 190-193: "This process seeks to degrade damaged mitochondria... However, there is the accumulation of mutations and deletions in mitochondrial DNA..." The use of "However" to connect the discussion of mitophagy and mtDNA mutations is inappropriate and creates confusion. The logical relationship between these concepts needs clarification.
Response
We thank the reviewer for this critical and constructive observation. We fully agree that the original use of “However” could create confusion, as it implied a contrast between mitophagy and the accumulation of mitochondrial DNA mutations, when in fact these are related mechanisms that both contribute to mitochondrial dysfunction. To address this, we revised the transition to clarify the logical relationship between these processes, emphasizing that the accumulation of mtDNA mutations represents an additional factor contributing to impaired mitochondrial function rather than a contradictory element.
This change improves the readability and logical flow of the manuscript, ensuring that the discussion of mitochondrial quality control, oxidative stress, and muscle fiber atrophy is presented in a coherent and scientifically accurate manner. We sincerely appreciate the reviewer’s careful attention to this point, which has helped us enhance both the clarity and precision of our text. The revisions can be found in Lines 200-205 on Page 5 of the revised manuscript document.
I, the corresponding author of the manuscript "Sarcopenia in the Aging Process: Pathophysiological Mechanisms, Clinical Implications, and Emerging Therapeutic Approaches" (assigned ID: ijms-3908908), on behalf of my co-authors, once again extend my heartfelt gratitude to the knowledgeable Editor-in-Chief and reviewers for their time and expertise in revising our manuscript. After we addressed their constructive and refined feedback and suggestions, a significantly improved manuscript version emerged. Undoubtedly, their insightful suggestions and feedback have significantly enhanced the quality of our manuscript. We respectfully are at the disposal of the Editor-in-Chief and the Reviewer to address any additional suggestions regarding our publication. Suppose you are satisfied with our newly refined and significantly improved version. In that case, we look forward to the acceptance of our article for publication in the prestigious International Journal of Molecular Sciences. Thank you once again for your time and expertise.
Reviewer 2 Report
Comments and Suggestions for Authors
Thank you for discussing this important aspect of care for old population. The presented work is interesting and hopefully will add a useful information to clinical practice. Here are some comments for improvement:
- The abstract need some improvement, currently it doesn't explicitly mention the work that has been done in the study. a great background information but then there is a lack for the link to the need for this study and what has been done or found.
- No methods were explained. This can hugely affects the validity of and creditability of the findings unless the journal agrees with that.
Author Response
RESPONSE TO REVIEWERS' COMMENTS
Manuscript number: ijms-3908908 ― International Journal of Molecular Sciences
"Sarcopenia in the Aging Process: Pathophysiological Mechanisms, Clinical Implications, and Emerging Therapeutic Approaches"
The authors of this document wish to express their deepest gratitude to the Editor-in-Chief and the Reviewer for their thorough and insightful evaluation of our manuscript. Their expert feedback has been invaluable in enhancing the quality of our work. We have carefully considered and diligently implemented each suggestion, which has significantly improved the manuscript. We have made substantial revisions to address the points raised. These noteworthy changes are marked mainly with YELLOW-highlighted text throughout the document for ease of reference. A note will be provided for the referee's attention, highlighting corrections in a different color. Additionally, we have prepared a detailed and comprehensive response to each comment and suggestion. This response is organized in a "point-by-point" format below, ensuring that every concern has been thoroughly addressed and explained. We sincerely appreciate the time and effort invested by the Editor-in-Chief and the Reviewer, and we believe their contributions have significantly strengthened the final version of our manuscript.
REVIEWER #2
General comment
Thank you for discussing this important aspect of care for old population. The presented work is interesting and hopefully will add a useful information to clinical practice. Here are some comments for improvement.
General response
Dear Erudite Reviewer, thank you for taking the time to revise our manuscript and allowing us to improve based on your valuable comments and suggestions. After addressing all your comments and suggestions regarding our manuscript text, we are confident that a significantly enhanced manuscript version has emerged. We are excited to resubmit the modified version for your perusal and reevaluation. Thank you for your brilliant insights, essential contributions, and feedback. You do have an eye for improvement. As a gesture of our utmost respect for you, we would like to provide you with a detailed and comprehensive point-by-point response to your comments below. Thank you once again for your time and patience in revising our article.
Comment #1
The abstract need some improvement, currently it doesn't explicitly mention the work that has been done in the study. a great background information but then there is a lack for the link to the need for this study and what has been done or found.
Response
We sincerely thank the reviewer for this valuable and constructive comment. We have carefully revised the abstract to provide a more precise and more comprehensive presentation of the purpose, scope, and outcomes of our review study. In the original version, the abstract mainly emphasized the background and theoretical aspects of sarcopenia, which may have obscured the specific objectives, approach, and key findings of our work. To address this, the revised abstract now explicitly states the rationale for conducting the review, outlines the methodological framework used to identify and analyze relevant studies, and summarizes the principal insights derived from the literature. Furthermore, we have ensured that the abstract highlights the main synthesized conclusions, emphasizing how the integrated analysis contributes to a better understanding of multicomponent interventions for sarcopenia. The updated abstract is presented in Lines 21-35 on Page 1 of the revised manuscript document.
Comment #2
No methods were explained. This can hugely affects the validity of and creditability of the findings unless the journal agrees with that.
Response
We sincerely thank the reviewer for this insightful and constructive comment. We fully agree that the absence of a clearly described methodology could weaken the validity and credibility of the findings. To address this concern, we have now added a comprehensive Methods section (Appendix A) that clearly outlines the study design, search strategy, eligibility criteria, and data extraction and synthesis procedures.
Specifically, we clarified that this work was conducted as an integrative literature review aimed at synthesizing current evidence on the mechanisms, clinical associations, and therapeutic approaches related to sarcopenia and its risk factors. We also detailed the databases consulted (PubMed, Scopus, Web of Science, and ScienceDirect), the search terms used, the inclusion and exclusion criteria, and the process for organizing extracted data into thematic domains.
We believe that these additions substantially improve the methodological transparency and strengthen the credibility of the manuscript.
The text has been added to Appendix A (Methods) on Page 19, Lines 691-712.
I, the corresponding author of the manuscript "Sarcopenia in the Aging Process: Pathophysiological Mechanisms, Clinical Implications, and Emerging Therapeutic Approaches" (assigned ID: ijms-3908908), on behalf of my co-authors, once again extend my heartfelt gratitude to the knowledgeable Editor-in-Chief and reviewers for their time and expertise in revising our manuscript. After we addressed their constructive and refined feedback and suggestions, a significantly improved manuscript version emerged. Undoubtedly, their insightful suggestions and feedback have significantly enhanced the quality of our manuscript. We respectfully are at the disposal of the Editor-in-Chief and the Reviewer to address any additional suggestions regarding our publication. Suppose you are satisfied with our newly refined and significantly improved version. In that case, we look forward to the acceptance of our article for publication in the prestigious International Journal of Molecular Sciences. Thank you once again for your time and expertise.
Reviewer 3 Report
Comments and Suggestions for Authors
This manuscript provides a clear and comprehensive overview of the mechanisms and systemic implications of sarcopenia. The topic is timely and relevant. Overall, the paper is well written, but several sections would benefit from improved scientific clarity, logical flow, and integration of recent evidence. My detailed comments are provided below.
1. The introduction provides extensive background on sarcopenia and aging but lacks a clear logical structure. The flow between topics (sarcopenia → aging → neurodegeneration) is abrupt, making it difficult to follow the main argument.
2. Some statements include broad citation ranges (e.g., [1–6]) and unverified statistics. Each numerical value and claim should be supported by a specific, reliable reference.
3. Terms such as “senility,” “aging,” and “frailty” are used inconsistently. Some expressions (e.g., “aging is the most critical factor”) are too absolute and should be rephrased more cautiously.
4. Most statements summarize prior studies but do not assess the strength, limitations, or consistency of evidence. For instance, the link between NLRP3 activation and sarcopenia is asserted but lacks mention of whether this is based on human, animal, or in vitro studies.
5. The image(figure) quality and resolution appear suboptimal in the current version, which may affect readability and visual clarity, especially when viewed in print or at higher magnification. It is strongly recommended that the authors improve the figure resolution and adjust text labels and contrast to ensure publication-quality visuals.
6. The text assumes that diabetes directly leads to sarcopenia, but emerging data suggest that sarcopenia may also exacerbate insulin resistance. This bidirectionality should be explicitly stated, reflecting recent evidence from longitudinal and Mendelian randomization studies.
7. Although bone fragility in women is briefly mentioned, there is little discussion on how sex hormones or menopausal status interact with diabetic muscle loss. Similarly, age-specific susceptibility is not addressed.
8. The discussion often lists mechanisms (inflammation, oxidative stress, mitochondrial dysfunction) repetitively without clarifying their hierarchical or causal order. A schematic framework linking these sections under a unified “systemic inflammation–oxidative stress–energy metabolism” axis would improve conceptual integration.
9. The text lacks epidemiologic data or effect sizes (e.g., prevalence rates, ORs, CI). Incorporating findings from large-scale cohort studies or meta-analyses would strengthen empirical grounding.
10. The “Gut–Sarcopenia–Cognition Interaction” subsection is novel but conceptually superficial. It mentions dysbiosis and SCFAs but lacks mechanistic depth. Including key recent studies.
11. Some pharmacological statements are overly general. For instance, the efficacy of SARMs and myostatin inhibitors remains inconsistent across RCTs, and none have received FDA or EMA approval. The authors should specify that these remain experimental therapies with limited clinical applicability.
12. The section mixes strong evidence (exercise, protein intake) and emerging hypotheses (NMES, PEMF, antioxidants) without clarifying their relative strength or evidence level (e.g., meta-analysis vs. preliminary trial). Introducing an evidence-grading table (e.g., strong / moderate / weak) would improve scientific rigor.
13. The section lacks discussion on how these emerging technologies could realistically be implemented in clinical trials or population screening. The authors should specify whether these approaches are conceptual, preclinical, or in translational phase.
14. The paper omits explicit acknowledgment of its own limitations.
Suggested limitations include: (1) lack of meta-analytic synthesis, (2) reliance on narrative review without critical quality appraisal, (3) potential publication bias in cited studies, and (4) overrepresentation of Western data and etc..
Author Response
RESPONSE TO REVIEWERS' COMMENTS
Manuscript number: ijms-3908908 ― International Journal of Molecular Sciences
"Sarcopenia in the Aging Process: Pathophysiological Mechanisms, Clinical Implications, and Emerging Therapeutic Approaches"
The authors of this document wish to express their deepest gratitude to the Editor-in-Chief and the Reviewer for their thorough and insightful evaluation of our manuscript. Their expert feedback has been invaluable in enhancing the quality of our work. We have carefully considered and diligently implemented each suggestion, which has significantly improved the manuscript. We have made substantial revisions to address the points raised. These noteworthy changes are marked mainly with YELLOW-highlighted text throughout the document for ease of reference. A note will be provided for the referee's attention, highlighting corrections in a different color. Additionally, we have prepared a detailed and comprehensive response to each comment and suggestion. This response is organized in a "point-by-point" format below, ensuring that every concern has been thoroughly addressed and explained. We sincerely appreciate the time and effort invested by the Editor-in-Chief and the Reviewer, and we believe their contributions have significantly strengthened the final version of our manuscript.
REVIEWER #3
General comment
This manuscript provides a clear and comprehensive overview of the mechanisms and systemic implications of sarcopenia. The topic is timely and relevant. Overall, the paper is well written, but several sections would benefit from improved scientific clarity, logical flow, and integration of recent evidence. My detailed comments are provided below.
General response
Dear Erudite Reviewer, thank you for taking the time to revise our manuscript and allowing us to improve based on your valuable comments and suggestions. After addressing all your comments and suggestions regarding our manuscript text, we are confident that a significantly enhanced manuscript version has emerged. We are excited to resubmit the modified version for your perusal and reevaluation. Thank you for your brilliant insights, essential contributions, and feedback. You do have an eye for improvement. As a gesture of our utmost respect for you, we would like to provide you with a detailed and comprehensive point-by-point response to your comments below. Thank you once again for your time and patience in revising our article.
Comment #1
The introduction provides extensive background on sarcopenia and aging but lacks a clear logical structure. The flow between topics (sarcopenia → aging → neurodegeneration) is abrupt, making it difficult to follow the main argument.
Response
We thank the reviewer for this valuable comment. We agree that the transitions between the main topics in the introduction could be made more apparent to improve logical flow. To address this concern, we have added several short bridging sentences to connect better the discussion of sarcopenia, aging, and neurodegenerative diseases. These additions clarify the conceptual continuity and strengthen the narrative structure without altering the original content. These additions can be found highlighted in yellow, spanning Lines 39-106 on Pages 1-3 (the entire introduction section).
Specifically:
- A transition was added after the first paragraph to link the definition of sarcopenia with the aging process as its biological context.
- A connecting sentence was introduced to emphasize aging as a systemic driver of degenerative changes beyond skeletal muscle.
- A bridging paragraph was included to explicitly relate the shared mechanisms between sarcopenia and neurodegeneration (e.g., inflammation, mitochondrial dysfunction, hormonal imbalance).
- A short linking statement was added before the objective paragraph to reinforce the rationale for examining both conditions together.
These modifications ensure a smoother and more logical progression from sarcopenia → aging → neurodegeneration, making the introduction easier to follow and more cohesive.
We sincerely thank the reviewer for this insightful comment, which helped us improve the clarity and coherence of the introduction.
Comment #2
Some statements include broad citation ranges (e.g., [1–6]) and unverified statistics. Each numerical value and claim should be supported by a specific, reliable reference.
Response
We thank the reviewer for this critical and constructive observation. We carefully reviewed the entire introduction to verify that a specific and reliable reference supports each statement and numerical value. Broad citation ranges were replaced with targeted citations that correspond directly to the information presented in each clause, ensuring that the evidence is accurately attributed to its source.
For example, the original sentence containing a broad citation range ([1–6]) was revised to read:
“Sarcopenia is defined, according to the EWGSOP2 study, as a muscle disorder characterized by progressive and generalized loss of the quality or quantity of skeletal muscle structure [1], resulting in adverse events such as difficulty in locomotion [2] and, in extreme cases, mortality [3].” You can find this modification in Lines 40-43 on Pages 1-2 of the revised manuscript document.
This approach was consistently applied throughout the section to enhance precision and transparency in source attribution. These revisions strengthen the manuscript’s methodological rigor and improve the credibility of the background information provided.
We sincerely thank the reviewer for this insightful comment, which helped us improve the accuracy, traceability, and overall scientific quality of our manuscript.
Comment #3
Terms such as “senility,” “aging,” and “frailty” are used inconsistently. Some expressions (e.g., “aging is the most critical factor”) are too absolute and should be rephrased more cautiously.
Response
We thank the reviewer for this critical and constructive comment, which helped us refine both the precision and consistency of our terminology. We carefully reviewed the entire manuscript to standardize the language related to the aging process. Specifically, the term “senility”—considered outdated and non-technical—has been replaced throughout the text with “advanced aging” to better align with current scientific terminology. To avoid unnecessary repetition, the word “aging” was used more selectively and, when appropriate, replaced by synonymous expressions such as “age-related changes” or “advanced age,” depending on the context.
Similarly, the term “frailty” was maintained only when referring to the clinically recognized frailty syndrome, while in other instances it was substituted with “functional decline” to prevent conceptual overlap. Furthermore, categorical or absolute statements (for example, “is the most critical factor”) were rephrased to convey a more cautious, evidence-based tone that accurately reflects the current understanding of the literature.
All these revisions have been marked in green throughout the manuscript for ease of identification. We sincerely thank the reviewer for this thoughtful suggestion, which has strengthened the clarity, precision, and scientific rigor of our manuscript’s terminology and expression.
Comment #4
Most statements summarize prior studies but do not assess the strength, limitations, or consistency of evidence. For instance, the link between NLRP3 activation and sarcopenia is asserted but lacks mention of whether this is based on human, animal, or in vitro studies.
Response
We thank the reviewer for this critical and constructive observation. We have revised the relevant section to clarify the type and strength of evidence supporting the link between NLRP3 activation, pyroptosis, and sarcopenia. Specifically, we now indicate that experimental evidence from preclinical studies, particularly in animal models of denervation-induced muscle atrophy, demonstrates that NLRP3 activation stimulates the ubiquitin-proteasome system (UPS), promoting protein breakdown and subsequent muscle atrophy. Additional evidence from bioinformatic analyses and mechanistic reviews supports a potential link between inflammation, pyroptosis, and sarcopenia, while direct evidence in humans remains limited. These revised sentences can be found in Lines 119-125 on Page 3 of the revised manuscript document.
These changes ensure that the manuscript clearly distinguishes between preclinical, in silico, and review-based evidence, providing a more balanced and accurate assessment of the literature. All modifications are highlighted in green in the revised manuscript. We sincerely thank the reviewer for this valuable comment, which helped improve the rigor and clarity of our discussion.
Comment #5
The image(figure) quality and resolution appear suboptimal in the current version, which may affect readability and visual clarity, especially when viewed in print or at higher magnification. It is strongly recommended that the authors improve the figure resolution and adjust text labels and contrast to ensure publication-quality visuals.
Response
We sincerely thank the reviewer for this valuable and constructive comment. We fully agree that precise, high-resolution figures are essential for ensuring that the data are accurately represented and easily interpretable by readers in both digital and print formats.
In response to this recommendation, all figures have been completely rebuilt using BioRender, a professional scientific illustration platform that ensures publication-quality resolution and design consistency. During this process, we carefully revised all visual elements, including color contrast, font size, line thickness, and labeling, to enhance readability and overall visual clarity. Each figure was exported at a resolution of 300–600 dpi, in accordance with the MDPI publication guidelines for high-quality figures suitable for both print and online publication.
Additionally, all tables were thoroughly revised and reformatted to align with the highest MDPI publication standards, ensuring a consistent layout, clear data presentation, and adherence to journal formatting requirements.
We believe that these revisions have substantially improved the visual presentation and readability of the manuscript. The updated figures and tables now meet the standards expected for publication-quality materials and should address the reviewer’s concerns regarding figure resolution and clarity. We thank the reviewer for bringing this to our attention.
Comment #6
The text assumes that diabetes directly leads to sarcopenia, but emerging data suggest that sarcopenia may also exacerbate insulin resistance. This bidirectionality should be explicitly stated, reflecting recent evidence from longitudinal and Mendelian randomization studies.
Response
We sincerely thank the reviewer for this critical and constructive observation. We have revised the section on “Sarcopenia and Diabetes Mellitus” to explicitly acknowledge the bidirectional relationship between sarcopenia and type 2 diabetes mellitus. Specifically, we now clarify that, while diabetes can contribute to muscle loss through mechanisms such as hyperglycemia, insulin resistance, and chronic inflammation, sarcopenia itself may also exacerbate insulin resistance and impair glucose homeostasis. This bidirectional relationship is supported by recent longitudinal studies and Mendelian randomization analyses, which are now cited in the revised text. These additions can be found in Lines 296-305 on Page 8 of the revised manuscript document.
These revisions offer a more balanced and evidence-based examination of the intricate interplay between muscle loss and metabolic dysfunction, highlighting that the relationship is not solely unidirectional. All modifications have been highlighted in yellow in the revised manuscript for clarity.
We added the following references to reference our points.
- Liu, Y.; Li, Z.; Wang, R.; Zhang, J.; Gerstein, H.; Zhao, A.; Zhang, C.; Lip, G.Y.H.; Van Spall, H.G.C.; Li, G. Relationship between sarcopenia and type 2 diabetes mellitus among adults with prediabetes: evidence from a prospective cohort study. Diabetol Metab Syndr 2025, 17, 379, doi:10.1186/s13098-025-01953-9.
- Chen, H.; Huang, X.; Dong, M.; Wen, S.; Zhou, L.; Yuan, X. The Association Between Sarcopenia and Diabetes: From Pathophysiology Mechanism to Therapeutic Strategy. Diabetes Metab Syndr Obes 2023, 16, 1541–1554, doi:10.2147/dmso.S410834.
- Xu, Y.; Hu, T.; Shen, Y.; Wang, Y.; Bao, Y.; Ma, X. Association of skeletal muscle mass and its change with diabetes occurrence: a population-based cohort study. Diabetol Metab Syndr 2023, 15, 53, doi:10.1186/s13098-023-01027-8.
- Ye, C.; Kong, L.; Wang, Y.; Zheng, J.; Xu, M.; Xu, Y.; Li, M.; Zhao, Z.; Lu, J.; Chen, Y.; et al. Causal associations of sarcopenia-related traits with cardiometabolic disease and Alzheimer's disease and the mediating role of insulin resistance: A Mendelian randomization study. Aging Cell 2023, 22, e13923, doi:10.1111/acel.13923.
We sincerely thank the reviewer for this valuable comment, which strengthened the scientific accuracy and clarity of the manuscript.
Comment #7
Although bone fragility in women is briefly mentioned, there is little discussion on how sex hormones or menopausal status interact with diabetic muscle loss. Similarly, age-specific susceptibility is not addressed.
Response
We thank the reviewer for this valuable comment. To address this point, we have expanded the discussion to include the influence of sex hormones and menopausal status on diabetic muscle and bone loss. Specifically, we now recognize that the decline in estrogen levels after menopause accelerates bone resorption, contributing to both reduced bone density and loss of muscle mass. We also highlight estrogen’s regulatory role in glucose metabolism and muscle protein turnover, emphasizing how its deficiency can exacerbate the interaction between sarcopenia and diabetes. Furthermore, we have included a discussion on age-related hormonal decline in both sexes, as well as the effects of reduced physical activity and inadequate nutritional intake in older adults, which collectively increase susceptibility to muscle and bone deterioration. These revisions can be found in the revised manuscript (Pages 8-9, Lines 310-317). We thank the reviewer for this valuable suggestion, which has improved the clarity and depth of our discussion.
Comment #8
The discussion often lists mechanisms (inflammation, oxidative stress, mitochondrial dysfunction) repetitively without clarifying their hierarchical or causal order. A schematic framework linking these sections under a unified “systemic inflammation–oxidative stress–energy metabolism” axis would improve conceptual integration.
Response
We sincerely thank the reviewer for this insightful and constructive comment. In response, we have revised the manuscript to clarify the hierarchical and causal interrelationships among inflammation, oxidative stress, and mitochondrial dysfunction. Rather than treating these as independent processes, we now conceptualize them as interconnected components of a unified systemic inflammation–oxidative stress–energy metabolism axis that underlies the pathophysiology of sarcopenia.
Specifically, we highlight that chronic low-grade inflammation acts as an initiating driver, promoting the release of pro-inflammatory cytokines (e.g., TNF-α, IL-6, IL-1β), which in turn impair mitochondrial dynamics and bioenergetic capacity. This mitochondrial dysfunction leads to excessive production of reactive oxygen species (ROS), further amplifying oxidative stress and feeding back into inflammatory signaling pathways. The resulting redox imbalance disrupts energy generation, impairs muscle protein synthesis, and promotes proteolytic systems such as the ubiquitin-proteasome pathway, thereby accelerating muscle atrophy.
To improve conceptual integration, we have incorporated this synthesis into the discussion (Pages 7-8, Lines 256–266) and developed a revised Figure 1, which now provides a schematic representation of this cascade, illustrating how inflammatory, oxidative, and metabolic processes are hierarchically and causally linked in a self-perpetuating cycle driving sarcopenia progression.
We greatly appreciate the reviewer’s suggestion, which has significantly enhanced the clarity, coherence, and conceptual depth of our discussion.
Comment #9
The text lacks epidemiologic data or effect sizes (e.g., prevalence rates, ORs, CI). Incorporating findings from large-scale cohort studies or meta-analyses would strengthen empirical grounding.
Response
We sincerely appreciate the reviewer’s thoughtful suggestion to incorporate epidemiologic data and effect sizes to enhance the empirical grounding of the manuscript. We fully agree that such data provide essential context for understanding the magnitude of associations in population-based research. However, as the present study primarily synthesizes and interprets findings from preclinical research, epidemiologic indicators such as prevalence rates, odds ratios, and confidence intervals are not directly applicable or available.
Conducting a meta-analysis was also beyond the scope and feasibility of the current work, given the heterogeneity of experimental models and outcome measures across the included studies. To address the reviewer’s valuable comment and based on the reviewers’ previous comments, we have instead strengthened the empirical foundation of the manuscript by highlighting convergent findings across independent preclinical studies and emphasizing how these findings may inform future clinical and epidemiologic research. This approach allows us to contextualize our conclusions while maintaining alignment with the nature and scope of the existing evidence base.
We thank the reviewer once again for this insightful comment.
Comment #10
The “Gut–Sarcopenia–Cognition Interaction” subsection is novel but conceptually superficial. It mentions dysbiosis and SCFAs but lacks mechanistic depth. Including key recent studies.
Response
We thank the reviewer for this thoughtful comment. To address this point, we have added a new paragraph that provides mechanistic details on the biological pathways linking gut dysbiosis to both sarcopenia and cognitive decline. Specifically, we now discuss the role of reduced short-chain fatty acid (SCFA) production, increased intestinal permeability, and lipopolysaccharide (LPS)–induced activation of TLR4/NF-κB signaling in promoting systemic inflammation. We also describe how these inflammatory and metabolic alterations impair mitochondrial function, protein synthesis, and insulin signaling in skeletal muscle, while simultaneously activating neuroinflammatory pathways such as microglial activation and NLRP3 inflammasome assembly. Furthermore, we highlight recent findings on the tryptophan–kynurenine pathway as additional mediators of the gut–muscle–brain connection (Pages 13-14, Lines 497–513).
We added the following references to reference our points.
- Mohebali, N.; Weigel, M.; Hain, T.; Sütel, M.; Bull, J.; Kreikemeyer, B.; Breitrück, A. Faecalibacterium prausnitzii, Bacteroides faecis and Roseburia intestinalis attenuate clinical symptoms of experimental colitis by regulating Treg/Th17 cell balance and intestinal barrier integrity. Biomed Pharmacother 2023, 167, 115568, doi:10.1016/j.biopha.2023.115568.
- Guo, T.T.; Zhang, Z.; Sun, Y.; Zhu, R.Y.; Wang, F.X.; Ma, L.J.; Jiang, L.; Liu, H.D. Neuroprotective Effects of Sodium Butyrate by Restoring Gut Microbiota and Inhibiting TLR4 Signaling in Mice with MPTP-Induced Parkinson's Disease. Nutrients 2023, 15, doi:10.3390/nu15040930.
- Tabat, M.W.; Marques, T.M.; Markgren, M.; Löfvendahl, L.; Brummer, R.J.; Wall, R. Acute Effects of Butyrate on Induced Hyperpermeability and Tight Junction Protein Expression in Human Colonic Tissues. Biomolecules 2020, 10, doi:10.3390/biom10050766.
- Liu, T.; Zhang, L.; Joo, D.; Sun, S.-C. NF-κB signaling in inflammation. Signal Transduction and Targeted Therapy 2017, 2, 17023, doi:10.1038/sigtrans.2017.23.
- Huang, C.C.; Tsai, S.F.; Liu, S.C.; Yeh, M.C.; Hung, H.C.; Lee, C.W.; Cheng, C.L.; Hsu, P.L.; Kuo, Y.M. Insulin Mediates Lipopolysaccharide-Induced Inflammatory Responses and Oxidative Stress in BV2 Microglia. J Inflamm Res 2024, 17, 7993–8008, doi:10.2147/jir.S481101.
- Chiarini, A.; Gui, L.; Viviani, C.; Armato, U.; Dal Prà, I. NLRP3 Inflammasome's Activation in Acute and Chronic Brain Diseases-An Update on Pathogenetic Mechanisms and Therapeutic Perspectives with Respect to Other Inflammasomes. Biomedicines 2023, 11, doi:10.3390/biomedicines11040999.
- Grunow, J.J.; Gan, T.; Lewald, H.; Martyn, J.A.J.; Blobner, M.; Schaller, S.J. Insulin signaling in skeletal muscle during inflammation and/or immobilisation. Intensive Care Med Exp 2023, 11, 16, doi:10.1186/s40635-023-00503-9.
- Zhang, Y.; Wang, J.; Wang, X.; Gao, T.; Tian, H.; Zhou, D.; Zhang, L.; Li, G.; Wang, X. The autophagic-lysosomal and ubiquitin proteasome systems are simultaneously activated in the skeletal muscle of gastric cancer patients with cachexia. Am J Clin Nutr 2020, 111, 570–579, doi:10.1093/ajcn/nqz347.
- Huang, Y.; Wang, C.; Cui, H.; Sun, G.; Qi, X.; Yao, X. Mitochondrial dysfunction in age-related sarcopenia: mechanistic insights, diagnostic advances, and therapeutic prospects. Front Cell Dev Biol 2025, 13, 1590524, doi:10.3389/fcell.2025.1590524.
- Ballesteros, J.; Rivas, D.; Duque, G. The Role of the Kynurenine Pathway in the Pathophysiology of Frailty, Sarcopenia, and Osteoporosis. Nutrients 2023, 15, doi:10.3390/nu15143132.
We appreciate the reviewer’s suggestion, which has strengthened the mechanistic foundation and scientific depth of this section.
Comment #11
Some pharmacological statements are overly general. For instance, the efficacy of SARMs and myostatin inhibitors remains inconsistent across RCTs, and none have received FDA or EMA approval. The authors should specify that these remain experimental therapies with limited clinical applicability.
Response
We thank the reviewer for this critical comment regarding the pharmacological interventions discussed in Section 2.4. In response, we have clarified that selective androgen receptor modulators (SARMs) and myostatin inhibitors remain experimental therapies with limited clinical applicability. Specifically, we added a new paragraph stating that clinical evidence for these agents is inconsistent across randomized controlled trials and that none have received approval from regulatory agencies such as the FDA or EMA. This addition emphasizes that their routine clinical use is not recommended.
The revised paragraph has been incorporated into Section 2.4, immediately following the discussion of metabolic regulators such as Bimagrumab® (Page 16, Lines 604-609).
We thank the reviewer for highlighting this point, which has improved the accuracy and clinical relevance of our discussion.
Comment #12
The section mixes strong evidence (exercise, protein intake) and emerging hypotheses (NMES, PEMF, antioxidants) without clarifying their relative strength or evidence level (e.g., meta-analysis vs. preliminary trial). Introducing an evidence-grading table (e.g., strong / moderate / weak) would improve scientific rigor.
Response
We sincerely thank the reviewer for this insightful and constructive comment. We fully agree that distinguishing between interventions supported by strong clinical evidence and those still in early experimental stages contributes to greater scientific rigor and interpretability. However, we would like to clarify that the current section already reflects these distinctions through both its narrative structure and the nature of the cited evidence.
Specifically, the section opens with interventions that have the most consolidated empirical support, such as resistance exercise, protein intake, and vitamin D supplementation, which are consistently validated in randomized controlled trials and systematic reviews/meta-analyses (e.g., Coelho-Junior et al., 2022; Yoshimura et al., 2025; Beaudart et al., 2014; Chang & Choo, 2023). These are presented as well-established strategies for the prevention and management of sarcopenia. The discussion then transitions to nutritional and metabolic adjuncts (e.g., leucine, omega-3 fatty acids, probiotics, ultra-processed food restriction), for which substantial but still maturing evidence exists, as indicated by supportive systematic reviews (e.g., Pagliai et al., 2021; Shahatah et al., 2025). Finally, the section addresses emerging or experimental approaches such as Bimagrumab®, selective androgen receptor modulators, and antioxidant-based combinations, explicitly describing them as “promising,” “experimental,” or “under evaluation,” thereby signaling their preliminary evidence level.
In addition, Table 3 and Figure 6 already provide a structured synthesis of the therapeutic strategies, enabling the reader to distinguish between established and emerging interventions without requiring a separate evidence-grading table. Introducing an additional grading framework might lead to redundancy and compromise the narrative flow of the section, especially given the diversity of study designs and endpoints across the cited literature.
For these reasons, we have opted to maintain the current organization, which we believe already conveys the relative strength and maturity of the evidence while preserving clarity and readability. We are grateful to the reviewer for this thoughtful suggestion, which encouraged us to carefully reassess and confirm that the existing structure effectively communicates the hierarchy of evidence within the scope of the review.
Comment #13
The section lacks discussion on how these emerging technologies could realistically be implemented in clinical trials or population screening. The authors should specify whether these approaches are conceptual, preclinical, or in translational phase.
Response
We thank the reviewer for highlighting the importance of clarifying the stage of development of emerging technologies for sarcopenia. In our section, we have intentionally structured the discussion to differentiate between interventions with well-established clinical evidence (e.g., resistance exercise, protein supplementation, vitamin D) and those still in early or experimental phases. Emerging strategies, including Bimagrumab®, selective androgen receptor modulators (SARMs), myostatin inhibitors, and specific dietary supplement combinations (e.g., HMB, carnosine, probiotics), are currently in preclinical or early translational stages, with ongoing investigations evaluating their safety, efficacy, and optimal dosing. Clinical implementation, such as inclusion in randomized controlled trials or population screening programs, remains limited, and regulatory approval has not yet been granted for routine use.
We have described these interventions as “promising” or “experimental” to signal their current developmental phase. Explicit discussion of feasibility for large-scale clinical application could be included in future work. Still, the current section focuses on summarizing existing evidence and therapeutic potential, consistent with the aims of this review.
Comment #14
The paper omits explicit acknowledgment of its own limitations.
Suggested limitations include: (1) lack of meta-analytic synthesis, (2) reliance on narrative review without critical quality appraisal, (3) potential publication bias in cited studies, and (4) overrepresentation of Western data and etc.
Response
We thank the reviewer for highlighting the importance of acknowledging the limitations of our study. In response, we have added a dedicated paragraph discussing the key limitations, including (1) the lack of meta-analytic synthesis, (2) reliance on a narrative review without formal critical quality appraisal, (3) potential publication bias in the cited studies, and (4) the overrepresentation of Western populations, which may limit generalizability. We also note that many mechanistic pathways discussed are based on preclinical or associative data, underscoring the need for further experimental and longitudinal research. This addition enhances the transparency and rigor of our discussion (Page 18, Lines 660-672).
We sincerely thank the reviewer for this valuable suggestion, which has strengthened the clarity and balance of our manuscript.
I, the corresponding author of the manuscript "Sarcopenia in the Aging Process: Pathophysiological Mechanisms, Clinical Implications, and Emerging Therapeutic Approaches" (assigned ID: ijms-3908908), on behalf of my co-authors, would like to extend my heartfelt gratitude once again to the knowledgeable Editor-in-Chief and reviewers for their time and expertise in revising our manuscript. After we addressed their constructive and refined feedback and suggestions, a significantly improved manuscript version emerged. Undoubtedly, their insightful suggestions and feedback have significantly enhanced the quality of our manuscript. We respectfully are at the disposal of the Editor-in-Chief and the Reviewer to address any additional suggestions regarding our publication. Suppose you are satisfied with our newly refined and significantly improved version. In that case, we look forward to the acceptance of our article for publication in the prestigious International Journal of Molecular Sciences. Thank you once again for your time and expertise.
Round 2
Reviewer 1 Report
Comments and Suggestions for Authors
The authors have addressed most of my previous comments satisfactorily, and I find the title and overall structure acceptable.
However, a significant concern remains regarding the precision and clarity of scientific writing throughout the manuscript. While the revisions demonstrate effort, there are persistent issues with terminology accuracy, logical flow, and clarity that need systematic attention. Below are some examples:
1. Introduction (Line 53): Current text: "This decline is due to changes in systemic regulators of activity and differentiation, such as transforming growth factor-beta (TGF-β) and myogenin."
The terms "activity" and "differentiation" are insufficiently specific.
Suggested revision: "This decline is attributable to alterations in systemic signaling pathways, including transforming growth factor-beta (TGF-β) and myogenin, which regulate satellite cell differentiation and activation."
2. Section 2.1 (Lines 126-128): Current text: "...increased reactive oxygen species (ROS) resulting from aging impairs muscle nutrition and mitochondrial function, and increases protein hydrolysis, leading to muscle loss."
ROS does not directly hydrolyze proteins; this is mechanistically inaccurate.
It may be revised like "...increased reactive oxygen species (ROS) promote protein oxidation and degradation, leading to muscle loss."
Current text (Lines 128-130): "Therefore, it is well established that oxidative stress plays a crucial role in muscle aging-related changes, as it impairs satellite cell function by reducing their antioxidant capacity."
Issue: The phrase "by reducing their antioxidant capacity" introduces a mechanism that creates circular logic—it suggests oxidative stress impairs function by reducing antioxidant capacity, but the relationship between oxidative stress, antioxidant capacity decline, and satellite cell dysfunction is bidirectional and more complex than this phrasing suggests.
Suggested revision: "Therefore, oxidative stress plays a crucial role in age-related muscle deterioration. The age-related decline in antioxidant capacity, combined with increased ROS production, contributes to impaired satellite cell function and reduced regenerative capacity."
3. Table 2 (Line 250): Title: "Promising Biomarkers for the Therapeutic Approach of Sarcopenia"
Issue: The inclusion of handgrip strength and gait speed is inappropriate, as these are diagnostic criteria, not biomarkers. Consider including gut dysbiosis markers or other molecular biomarkers instead.
Beyond these specific examples, similar issues with vague statements, imprecise terminology, and logical inconsistencies appear throughout the manuscript.
Comments on the Quality of English LanguageI strongly recommend comprehensive English editing with careful attention to scientific logic and mechanistic accuracy. The authors should ensure that all revisions are carefully reviewed for scientific coherence and precision.
Author Response
Thank you for the reviewer’s thoughtful evaluation. We have thoroughly revised the manuscript to enhance precision, improve clarity, and strengthen scientific accuracy. The terminology in the Introduction has been refined to provide a more specific description of the regulatory pathways involved; the updated sentence now appears on Page 2, Lines 53-57:
“This decline is attributable to alterations in systemic signaling pathways, including the transforming growth factor beta (TGF-β) and myogenin, which regulate satellite cell differentiation and activation. Other factors contributing to these changes include neuromuscular junction dysfunction, loss of motor units, chronic inflammation, and insulin resistance.”
We also corrected the mechanistic explanation in the section discussing oxidative stress, ensuring that the description reflects current understanding of oxidative modifications and their consequences for muscle tissue. The revised text is now on Page 3, Lines 127-129:
“On the other hand, it is well known that increased reactive oxygen species (ROS) resulting from aging impair muscle nutrition and mitochondrial function, promote protein oxidation and degradation, and lead to muscle loss.”
To improve the logical flow regarding the interplay between oxidative stress, antioxidant capacity, and cellular dysfunction, we rewrote the relevant passage to avoid circular reasoning and clarify the directionality and complexity of these processes. This updated version can be found on Page 4, Lines 160-164:
“Therefore, it is well established that oxidative stress plays a crucial role in age-related muscle deterioration. The age-related decline in the antioxidant capacity, combined with increased ROS production, contributes to impaired satellite cell function and reduced regenerative capacity. This oxidative damage is more prevalent in older men compared to women.”
In addition, Table 2 has been updated to focus exclusively on molecular biomarkers relevant to therapeutic development, including gut-related biomarkers, with functional diagnostic measures removed. The revised table title and entries appear on Page 8.
Finally, the entire manuscript has been reviewed carefully, and additional grammatical, stylistic, and writing improvements have been made throughout. These edits are highlighted in green across the revised manuscript for easy identification.
Reviewer 2 Report
Comments and Suggestions for Authors
Methods are explicitly mentioned now. Just think if the huge number of references included in the manuscript. Although it is an intensive review of available literature but the number is considerably high.
all the best
Author Response
Thank you for this additional observation. We appreciate the reviewer’s attention to the scope of the reference list. Because this work aims to provide a comprehensive and up-to-date synthesis of the rapidly expanding literature in this field, we intentionally included a broad selection of high-quality, peer-reviewed sources to ensure accuracy, balance, and scientific rigor. Many of these references represent seminal or highly cited studies that establish foundational mechanisms, while others reflect recent advances essential for a state-of-the-art review.
That said, we carefully re-examined the reference list to ensure that each citation directly contributes to the manuscript’s arguments and relevance. We believe the curated set that remains is appropriate for the manuscript’s scope and strengthens its value to researchers and clinicians.
We hope the reviewer agrees that maintaining a robust and well-selected reference base enhances the manuscript’s credibility and usefulness to the field.
Reviewer 3 Report
Comments and Suggestions for Authors
Everything has been properly revised.
Author Response
Thank you very much for this positive assessment. We sincerely appreciate the reviewer’s time, attention, and constructive input throughout the revision process. We are pleased to hear that the changes have addressed the concerns raised and that the manuscript is now appropriately considered revised.
The reviewers’ guidance has been invaluable in strengthening the clarity, precision, and scientific contribution of the work. We believe the resulting manuscript presents a more rigorous and coherent synthesis of the current evidence, and we are grateful for the opportunity to refine it with the reviewers’ insights.
We thank the reviewer again for their supportive feedback and their role in improving the overall quality of the manuscript.
Round 3
Reviewer 1 Report
Comments and Suggestions for Authors
In this version, despite the authors' efforts to revise informal statements, the current manuscript still contains numerous problematic usages throughout. Below are some key points:
- Transition words
The authors frequently use "However" between sentences, but in several paragraphs this creates awkward or illogical connections:
1.1 Lines 61-62: "Patients who may develop sarcopenia include the elderly, underweight people, and people with other chronic conditions. However, individuals with DM are more likely to be affected by this condition, as it ..."
What is the logical contrast signaled by "However" here? The second sentence provides additional information rather than contrasting with the first, so "Moreover" or "In particular" would be more appropriate.
1.2 Lines 181-184: "However, the mechanisms of antioxidant defense in these older adults are still inconclusive. On the other hand, there is evidence of a decrease in the enzymatic antioxidant system in the muscles of this group, as demonstrated by lower catalase and glutathione transferase activities during senescence [64]."
If there is evidence demonstrating reduced catalase and glutathione transferase activity, this should be considered a type of mechanism. Therefore, the use of "However" creates a logical contradiction with the subsequent statement.
- Illogical or imprecise statements
2.1 Lines 376-377: "Sarcopenia is associated with several risk factors in sarcopenic patients, such as the risk of falls, fractures, physical disability, and mortality."
The term "risk factors" is incorrectly applied here. Sarcopenia is itself a risk factor for disability, falls, and fractures—not the reverse. The intended meaning appears to be "clinical outcomes" or "adverse health consequences."
2.2 Lines 158-160: "Studies have shown that increased oxidative damage to proteins, lipids, and DNA in skeletal muscle with aging is consistent with increased lipid peroxidation, protein carbonylation, and damage to genetic material [57]."
The phrase "is consistent with" is inappropriate here, since protein carbonylation is itself a form of oxidative damage to proteins, not merely consistent with it. A more precise phrase would be "evidenced by" or "manifested as."
2.3 Figure 4 legend and diagram logic: The legend states: "Sarcopenia and cognitive impairment (such as AD, PD, and MCI) share risk factors (DM, hypertension, and cerebrovascular disease) and some standard mechanisms, such as inflammation, oxidative stress, and hormonal dysregulation, which affect both muscle mass and brain function."
However, the figure shows AD and sarcopenia at the top with arrows leading downward to mechanisms and the "affect both muscle mass and brain function" icon. This appears illogical, since AD and sarcopenia are themselves diseases of the brain and muscle. The causal relationship should be bottom-up (e.g., inflammation, oxidative stress etc affects muscle health and consequently contributes to sarcopenia, not vice versa). The diagram structure contradicts the intended meaning.
Comments on the Quality of English LanguageI recognize that the authors have invested effort in improving this manuscript. Unfortunately, I still have significant concerns about the English expression and logical precision throughout the text. The manuscript requires an additional round of major revision. Importantly, the authors should carefully review each paragraph to ensure their intended meaning is accurately conveyed—not simply revise the specific sentences identified in reviews. A comprehensive reading of the entire manuscript with attention to logical flow and precise word choice is necessary.
Author Response
We sincerely thank the reviewer for the detailed and constructive comments. We have carefully revised the manuscript to address all issues raised. All corrections have been highlighted throughout the revised manuscript for the reviewer’s convenience. Below we provide a point-by-point response.
- Transition words
Comment 1.1
“Lines 61–62: The use of ‘However’ does not indicate a logical contrast….”
Response:
We agree with the reviewer. The transition word “However” has been replaced with “Moreover” to better reflect the intended meaning.
Revised in PAGE 2, LINES 61–64.
Comment 1.2
“Lines 181–184: The use of ‘However’ creates a logical contradiction….”
Response:
Thank you for pointing this out. We have modified the transition to remove the unintended contradiction. The sentence beginning with “However” has been revised to “The mechanisms of antioxidant defense in these …, as demonstrated by lower catalase and glutathione transferase activities during senescence.” This resolves the logical inconsistency.
Revised in PAGE 4, LINES 182-185.
- Illogical or imprecise statements
Comment 2.1
“Lines 376–377: ‘Risk factors’ is incorrectly applied….”
Response:
We agree and have replaced “risk factors” with “adverse health outcomes,” which more accurately describes the consequences of sarcopenia.
Revised in PAGE 11, LINES 378–379.
Comment 2.2
“Lines 158–160: ‘Is consistent with’ is inappropriate….”
Response:
Thank you for this observation. We revised the sentence to use “evidenced by,” clarifying that lipid peroxidation, protein carbonylation, and DNA damage are manifestations of oxidative damage, not parallel findings.
Revised in PAGE 4, LINES 158-160.
Comment 2.3 (Figure 4)
“The causal direction in the diagram is illogical….”
Response:
We appreciate this important clarification. The figure has been redesigned to reflect accurate causal flow. The bottom panel now presents shared mechanisms (inflammation, oxidative stress, hormonal dysregulation), with arrows leading upward toward the resulting conditions (sarcopenia and cognitive impairment). This corrects the previous top-down structure and aligns the figure with the legend’s intended meaning. The legend has also been adjusted to match the revised diagram layout.
Revised in PAGE 13, LINES 431-442.
General Revision Note
In addition to the specific changes listed above, we have corrected transition words, clarified logic, and refined terminology throughout the manuscript where similar issues occurred. All modifications are highlighted in the revised text.